# Comparison of Single Doppler and Multiple Doppler Wind Retrievals in Hurricane Matthew (2016)

Ting-Yu Cha[1] and Michael M. Bell[1]

[1]Department of Atmospheric Science, Colorado State University, Fort Collins, Colorado, United States

**Correspondence:** Ting-Yu Cha (tingyu@colostate.edu)

**Abstract.** Hurricane Matthew (2016) was observed by the ground-based polarimetric Next Generation Weather Radar (NEXRAD) in Miami (KAMX) and NOAA P-3 airborne tail Doppler radar near the coast of the southeastern United States for several hours, providing a novel opportunity to evaluate and compare single and multiple Doppler wind retrieval techniques for tropical cyclone flows. The generalized velocity track display (GVTD) technique can retrieve a subset of the wind field from a single ground-based Doppler radar under the assumption of nearly axisymmetric rotational wind, but is shown to have errors from aliasing of unresolved wind components. An improved technique that mitigates errors due to storm motion is derived in this study, although some spatial aliasing remains due to limited information content from the single Doppler measurements. A spline-based variational wind retrieval technique called SAMURAI can retrieve the full three-dimensional wind field from airborne radar fore-aft pseudo-dual Doppler scanning, but is shown to have errors due to temporal aliasing from the non-simultaneous Doppler measurements. A comparison between the two techniques shows that the axisymmetric tangential winds are generally comparable between the two techniques, and the improved GVTD technique improves the accuracy of the retrieval. Fourier decomposition of asymmetric kinematic and convective structure shows more discrepancies due to spatial and temporal aliasing in the retrievals. The strengths and weaknesses of each technique for studying tropical cyclone structure are discussed, and suggest that complementary information can be retrieved from both single and dual Doppler retrievals. Future improvements to the asymmetric flow assumptions in single Doppler analysis and steady-state assumptions in pseudo-dual Doppler analysis are required to reconcile differences in retrieved tropical cyclone structure.

## 1 Introduction

Doppler radar can provide high-resolution wind measurements within tropical cyclones (TCs), but the measurement is limited to the projection of the wind along the radial direction of the radar beam. Wind retrieval techniques are therefore required in order to identify the convective and kinematic structure of TCs from either single or multiple Doppler observations. While multiple Doppler retrievals are generally superior for deriving three-dimensional winds, measurements from two or more radars are not generally available and are often not simultaneous. In addition to the presence of an airborne Doppler radar with fore/aft capability or multiple radars with sufficient range and geometry around the TC, a steady state assumption during the Doppler radar observation period is required to synthesize the wind fields into one snapshot in time. The steady state assumption is less severe for single Doppler wind retrievals, but more assumptions about the unresolved components of the

flow are required. Previous studies have shown the intercomparison of dual Doppler wind fields from two orthogonal flight legs and a ground-based two-radar network (Jorgensen et al., 1983; Hildebrand and Mueller, 1985). Several other studies have investigated both single and multi-Doppler techniques for retrieving TC wind fields (Lee et al., 1994; Crum et al., 1998; Reasor et al., 2000; Lee et al., 1999; Jou et al., 2008; Bell et al., 2012), but the strengths and weaknesses of different techniques have not been compared and addressed fully. In this study, ground-based single Doppler and airborne dual Doppler observations simultaneously sampling Hurricane Matthew (2016) are analyzed to provide the first comprehensive comparison between ground-based single and airborne multi-Doppler wind retrieval techniques in a TC.

One of the primary dual-Doppler platforms for TC studies is the airborne National Oceanic and Atmospheric Administration WP-3D (NOAA P-3) tail Doppler radar (TDR), which can obtain kinematic structure for storms well away from the U.S. coast. The TDR has been used since the early 1980's for airborne radar data collection in hurricane reconnaissance and research missions. In early usage, the wind field was reconstructed from using pseudo-dual Doppler analysis of two flight legs that were perpendicular to each other known as the "L" pattern (Marks and Houze, 1984, 1987). The "L" pattern takes a period of 0.5 to 1 hour to complete, such that some slowly evolving wind asymmetries can be deduced (Marks et al., 1992), but more rapidly evolving structure cannot be retrieved by this technique.

In Lee et al. (1994), the Velocity Track Display (VTD) technique was proposed to retrieve the TC kinematic structure from a single airborne Doppler radar. The VTD technique takes advantage of the Doppler signatures of a vortex with a dipole pattern of approaching and receding velocities. The linear least squares method is utilized to fit Fourier basis functions onto the Doppler velocity, such that a subset of the wind field can be retrieved for each radius and altitude based on the Fourier coefficients. VTD retrievals from Hurricane Gloria (1985) were compared with pseudo-dual Doppler analysis constructed from two orthogonal flight legs under the assumption that the circulation was in a steady state over 1-2 hour period. Lee et al. (1994) concluded that spatial and temporal aliasing in the pseudo-dual Doppler analysis over long periods tended to create artificial higher wavenumbers and reduce the wind maxima compared to the single Doppler retrievals. The study was one of the first to quantify the accuracy of low-wavenumber TC asymmetries from multiple Doppler analysis due to the evolving weather.

The "Fore/Aft Scanning Technique" (FAST) was proposed by Hildebrand et al. (1986) in which the antenna alternately scans forward and then aft of the flight track within a few seconds, which could produce more accurate local wind estimations. This scanning methodology can gather data faster, mitigating some of the impact from weather evolution, and no longer requires the execution an "L" shaped flight pattern (Gamache et al., 1995). The FAST approach produces horizontal winds with high accuracy, and can yield vertical motion through the mass continuity equation or a multi-beam technique with multiple aircraft (Jorgensen et al., 1996). The FAST scanning strategy has been used almost exclusively in recent years, and has provided significant advances of our understanding of TC structure and dynamics (Lee et al., 2003; Houze, 2010).

Although dual-Doppler observations can be used to assess snapshots of high resolution kinematic and convective structure, airborne reconnaissance and research missions are rare events in most of the countries impacted by TCs. The three-dimensional airflow structure can also be retrieved from the dual-Doppler observations when the system is detected by two ground-based radars. General sources of error in the inter-comparison of ground-based and airborne dual-Doppler observations include

instrument effects, algorithm effects, and sampling effects (Hildebrand and Mueller, 1985). Instrument effects include the effects of attenuation, signal-to-noise ratio, and the number of radar samples that could be caused by the radar processor design or measurement technique. These effects are likely to be most influential with marginal signal-to-noise, but random velocity errors up to 1 m s$^{-1}$ are possible with many radar designs, including airborne radars (Hildebrand et al., 1994). Algorithm effects include the effects of the interpolation to the Cartesian grid, multi-Doppler geometry, the solution method and its associated assumptions, and the derivation of the vertical velocities. Sampling effects include the effects of data spacing and density, geometry of flight tracks, temporal changes in the storm, advection, and data collection period. One of the long-lasting problems is the length of time required for each flight leg with airborne Doppler radar (Ray and Stephenson, 1990).The temporal effects can degrade the analysis if the data collection takes too long. Jorgensen et al. (1983) quantitatively compared the wind fields of homogeneous precipitation derived from the two pseudo-orthogonal flight legs and two ground-based dual-Doppler observations. Their measurements showed agreement in the horizontal wind fields, but small discrepancies in the vertical velocities about 0.5 - 1 m s$^{-1}$ for the airborne system and about 0.2 m s$^{-1}$ for the ground-based system. The discrepancy was attributed to uncertainties in the pointing angle of the airborne system and a long data collection period.

Ground-based dual-Doppler radar observations of TCs are usually limited to the observation of storms that happen to develop or move within the domain covered by the radars and extensive radar baselines Jou et al. (1996). The range limitation of ground-based radar observations can restrict the operational exploitation of dual Doppler measurements due to the large spacing between coastal radars and limited dual-Doppler lobes. As such, single Doppler retrieval techniques are often required to estimate the vortex structure.

The Ground-based velocity track display (GBVTD) was developed by Lee et al. (1999) as an extension of VTD for stationary radar scanning geometry. The GBVTD technique provides a new way to examine axisymmetric and asymmetric structures of a TC near landfall from single ground-based Doppler radar, and has been successfully utilized in several studies (Lee et al., 2000; Lee and Bell, 2007; Zhao et al., 2008, 2012; Shimada et al., 2018). One limitation is that the radial distance between the radar and the storm center has to be large enough to sample the tangential component of the vortex circulation in order to minimize the geometric distortion. Additionally, the GBVTD technique cannot fully separate the asymmetric components of the tangential and radial wind or the mean environmental wind due to spatial aliasing. Extensions to the GBVTD technique have been developed to better resolve asymmetries when multiple radars are available (EGBVTD) (Liou et al., 2006) or to resolve the mean wind when sufficient scatterers are near the radar (MGBVTD) (Chen et al., 2013).

The generalized velocity track display (GVTD) (Jou et al., 2008) is a technique that improves upon GBVTD by introducing an aspect ratio calculated by multiplying the distance of each gate ($D$) by measured Doppler velocity ($V_d$) and then scaling by the distance between the radar and the TC center ($R_T$). Key vortex kinematic structures displayed in the $V_d D/R_T$ space simplify the interpretation of the radar signature and eliminate the geometric distortion inherited in the $V_d$ space (Jou et al., 1996). GVTD expands $V_d D/R_T$ into Fourier coefficients in a linear coordinate ($\theta'$) rather than expanding $V_d$ in a nonlinear coordinate ($\psi'$) in GBVTD. The geometry and symbols of GVTD are displayed in Fig. 1. The retrieved wind field from GVTD is no longer limited by the analysis domain due to the required approximation of $\cos\alpha$ in GBVTD (Eq. (5) in Lee et al. (1999)), and the retrieved asymmetric structures are without distortion. The percentage errors of the retrieved wavenumber 2 and 3

asymmetries are negligible ($<1\%$) in general, which agrees well with the analytical solutions. Also, the GVTD formulation can be applied to the extensions of the velocity track display (VTD) techniques (e.g. GBVTD-simplex, Lee and Marks (2000)) to improve their performance. Therefore, it not only expands the capability of using ground-based Doppler radar data in TC forecasts but provides researchers an opportunity to examine TC kinematic and some derived dynamic variables in detail (such as vertical velocity, angular momentum, and vertical vorticity).

Jou et al. (2008) tested the GVTD technique with idealized vortices and confirmed that the modification by the aspect ratio is beneficial to retrieve the vortex kinematic structure. Although the GVTD technique largely improves and extends the capability of GBVTD, several limitations of the single-Doppler measurement remain. In particular, the equations are still underdetermined, and require a closure assumption in order to retrieve the asymmetric wind field.

The primary motivation of this study is to compare the strengths and weaknesses of the GVTD-technique and an airborne dual-Doppler wind-synthesis analysis in a real case, which has not been done in previous studies. Hurricane Matthew (2016) was observed by the Next Generation Weather Radar (NEXRAD) radar in Miami (KAMX), concurrently with the NOAA P-3 (hereafter P3) airborne radar when it approached the southeastern United States. The KAMX radar has a larger data coverage, but the P3 has better spatial resolution due to its closer range. Since airborne and ground-based radar collected data simultaneously, the case provides a unique opportunity to evaluate the wind retrievals. The datasets and methodology are described in section 2. The improved algorithm of the GVTD is formulated in section 3. In section 4, the improved GVTD algorithm is applied to the NEXRAD data in Hurricane Matthew (2016) and compared with the retrieved winds from the dual Doppler analysis. A summary of our results and conclusions are presented in section 5.

## 2    Datasets and methodology

Hurricane Matthew (2016) was the first Category 5 hurricane in the Atlantic basin since 2007, and caused widespread damage across its destructive path. When Matthew moved parallel to the east coast of Florida, it was observed simultaneously by the KAMX single Doppler radar at a 5-minute interval and the P3 TDR from 19 UTC 6 October to 00 UTC 7 October with four flight passes through the center during Matthew's eyewall replacement process.

The flight track of the P3 and the detecting range of the KAMX radar are displayed in Fig. 2a. All radar sweep files were initially processed using an automated quality control (QC) script using National Center for Atmospheric Research (NCAR) SoloII software (Bell et al., 2013) and then manually edited to unfold the Doppler velocity aliasing and remove the discontinuities and noise echoes. A coordinate transform and interpolation were applied to the KAMX radar fields from the original plan position indicators (PPIs) to the constant-altitude plan position indicators (CAPPIs) in Cartesian coordinates using Radx2Grid in the Lidar Radar Open Software Environment (LROSE) software (Bell, 2019). The gridded domain is 400 km $\times$ 400 km with a horizontal grid spacing of 1 km and vertical grid spacing of 0.5 km. While the vertical resolution may be a bit fine, we focus on the horizontal structure which is appropriately resolved for the given sampling. The gridded data was further analyzed using the Vortex Objective Radar Tracking and Circulation (VORTRAC) software in LROSE to interpolate onto a cylindrical coordinate and obtain the kinematic structure by the improved GVTD algorithm formulated in section 3.1.

The P3 was involved in the reconnaissance mission from 19 UTC 6 October to 00 UTC 7 October. The P3 was equipped with the TDR, which scanned in FAST mode in order to obtain pseudo dual-Doppler measurements. The TDR documented the intensification and weakening stages of Matthew's ERC. The P3 flew four radial passes through the center of the TC, with each pass being 30 to 60 minutes apart. The time window of the four passes are listed in Table 1, while the location of each pass is shown in Fig. 2a. The storm center and storm motion were both estimated from the Hurricane Research Division (HRD) aircraft-derived dynamic center (Willoughby and Chelmow, 1982). The analysis track for dual-Doppler analyses was linearly interpolated from each dynamic center using the derived storm motion. The dual-Doppler analysis was synthesized with each of the P3 radial passes at 1-km horizontal spline nodal spacing and 0.5 km vertical nodal spacing using SAMURAI software (Bell et al., 2012) in LROSE, with a $4\Delta x$ Gaussian filter in the horizontal and $2\Delta x$ filter in the vertical applied. SAMURAI is a three-dimensional variational data assimilation tool that uses a finite element basis to estimate the most likely state of the atmosphere given a set of observations. The nodal spacing of the finite elements should be smaller than the data spacing in order to accurately represent a spline function that can depict the spatial scales resolved by a given data sampling (e.g. Koch et al., 1983; Ooyama, 1987, 2002). For the P-3 TDR in 2016, the data spacing is limited in the along-track direction to $\sim$ 1.4 km due to the rotation rate of the radar. With the chosen spline nodal spacing and Gaussian filter length the minimum resolved scale is $\sim$4 km in the horizontal, or approximately 2.85 times the along-track data spacing. Larger-scale features such as low azimuthal wavenumber structures are well-resolved by the analysis. Additional algorithm effects of SAMURAI have been tested in Bell et al. (2012) which the analysis has high fidelity to observations with low noise, with linear correlations of 0.99 and linear slope and bias values near one and zero, respectively. The analysis was initially done on a Cartesian coordinate and then interpolated onto a cylindrical coordinate with azimuthal resolution of 1 degree and radial resolution of 1 km. The four passes were analyzed in detail to examine the changes in kinematic structure with high spatial resolution over four hours, with a particular focus on the first pass in this study.

## 3    The GVTD technique improvement

The Doppler velocity in Jou et al. (2008) is decomposed into tangential, radial, and mean wind components where storm motion is an implicit element in the mean wind component. The mean wind ($V_M$) is the horizontal average of the environmental flow at each altitude following the procedure proposed by Marks et al. (1992), which can be used to calculate the vertical wind shear. While the divergence of the environmental flow may not be zero since the environmental flow is not constant in the horizontal direction (Chan and Gray, 1982; Chan, 1984), we assume a horizontally homogeneous mean wind in the following derivation. This assumption is different than the original derivation in Jou et al. (2008) where the mean wind is a function of radius and height. Storm motion ($U_S$, $V_S$) is defined here as the deep layer motion vector over the whole vortex that varies only in time and does not vary with height or radius. The remaining terms in the GVTD formulation, namely the tangential ($V_T$)and radial ($V_R$) winds, are functions of radius, height and azimuth to the storm center. Despite the fact that the environmental wind is an important factor to determine storm motion, the storm motion and environmental wind are not the same component due to vertical wind shear.

The storm motion and mean wind component were originally combined together in GVTD and affected the retrievals of the mean wind, and axisymmetric (wavenumber 0) tangential and radial winds. Notably, they do not have an influence on retrieving the phase and magnitude of GVTD-asymmetric components of tangential wind. However, when the storm moves fast or there is a high deviation of direction between mean wind and storm motion they can produce errors. To resolve this error, we re-derive the GVTD technique and separate the storm motion from the mean wind component.

## 3.1 Mathematical formulation

Following the symbols and geometry utilized in the GBVTD (Lee et al., 1999) and GVTD (Jou et al., 2008) techniques, the addition of the storm motion to the geometry is illustrated in Fig. 1. We start with the horizontal projection of the Doppler velocity:

$$\hat{V}_d/cos\phi = V_M cos(\theta_d - \theta_M) - V_T sin\psi + V_R cos\psi \tag{1}$$

where $\phi$ is the elevation angle of the radar, $\theta_d$ is the mathematical angle of the radar measured from the east, $\theta_M$ is the direction of mean wind, and $\psi$ is the angle composed by the measured radar beam to the radar and the measured radar beam to the storm center. Note that $\hat{V}_d$ neglects the contribution from the terminal velocity ($v_t$) and vertical velocity ($w$) (Eq. 2 in Lee et al. (1999)). The contribution from $w$ and $v_t$ is small if the elevation angle of the radar beam is low ($<1°$). In this study, the storm motion ($U_S$, $V_S$) is added into the equation as an independent, known variable that projects onto the Doppler velocity, such that:

$$\hat{V}_d/cos\phi = V_M cos(\theta_d - \theta_M) + U_S cos\theta_d + V_S sin\theta_d - V_T sin\psi + V_R cos\psi \tag{2}$$

where $\psi = \theta - \theta_d$. Rearranging Eq. 2, we obtain

$$\begin{aligned}
\hat{V}_d/cos\phi = {} & V_M cos(\theta_d - \theta_M) - V_T sin(\theta - \theta_d) + V_R cos(\theta - \theta_d) + U_S cos\theta_d + V_S sin\theta_d \\
= {} & V_M(cos\theta_d cos\theta_M + sin\theta_d sin\theta_M) - V_T(sin\theta cos\theta_d - cos\theta sin\theta_d) \\
& + V_R(cos\theta cos\theta_d + sin\theta sin\theta_d) + U_S cos\theta_d + V_S sin\theta_d
\end{aligned} \tag{3}$$

The radar angle $\theta_d$ can be denoted as:

$$\begin{aligned}
D cos\theta_d = R cos\theta + R_T cos\theta_T \\
D sin\theta_d = R sin\theta + R_T sin\theta_T
\end{aligned} \tag{4}$$

Plugging Eq. 4 into Eq. 3 and approximating $\hat{V}_d/cos\phi$ with $V_d$ (only valid when the elevation angle is low):

$$\begin{aligned}
V_d = {} & (-V_T sin\theta + V_R cos\theta + V_M cos\theta_M + U_S)(R cos\theta + R_T cos\theta_T)/D \\
& + (V_T cos\theta + V_R sin\theta + V_M sin\theta_M + V_S)(R sin\theta + R_T sin\theta_T)/D
\end{aligned} \tag{5}$$

Rearranging Eq. 5 and let $\theta' = \theta - \theta_T$, we obtain:

$$V_d\frac{D}{R_T} = [V_R\frac{R}{R_T} + V_M cos(\theta_T - \theta_M) + U_S cos\theta_T + V_S sin\theta_T]$$
$$- [V_T + \frac{R}{R_T}(V_M sin(\theta_T - \theta_M) + U_S sin\theta_T - V_S cos\theta_T)]sin\theta'$$
$$+ [V_R + \frac{R}{R_T}(V_M cos(\theta_T - \theta_M) + U_S cos\theta_T + V_S sin\theta_T)]cos\theta'$$

(6)

Decomposing $V_d D/R_T$, $V_T$, and $V_R$ into Fourier components in the $\theta'$ coordinates:

$$V_d\frac{D}{R_T}(R,\theta') = A_0 + \Sigma A_n cosn\theta' + \Sigma B_n sinn\theta'$$

(7)

$$V_T(R,\theta') = V_T C_0 + \Sigma V_T C_n cosn\theta' + \Sigma V_T S_n sinn\theta'$$

(8)

$$V_R(R,\theta') = V_R C_0 + \Sigma V_R C_n cosn\theta' + \Sigma V_R S_n sinn\theta'$$

(9)

where $A_n$ ($V_T C_n$ and $V_R C_n$) and $B_n$ ($V_T S_n$ and $V_R S_n$) are the amplitude of the azimuthal wavenumber $n$ cosine and sine components, as defined in Lee et al. (1999); Jou et al. (2008). Substituting Eqs. 7, 8, and 9 into Eq. 6, we obtain the following expressions for the relation between the Fourier coeffients and wind components:

$$A_0 = \frac{R}{R_T}V_R C_0 + V_M cos(\theta_T - \theta_M) - \frac{1}{2}V_T S_1 + \frac{1}{2}V_R C_1 + U_S cos\theta_T + V_S sin\theta_T$$

(10)

$$A_1 = \frac{R}{R_T}V_R C_1 + \frac{R}{R_T}(V_M cos(\theta_T - \theta_M) + U_S cos\theta_T + V_S sin\theta_T) + V_R C_0 - \frac{1}{2}V_T S_2 + \frac{1}{2}V_R C_2$$

(11)

$$B_1 = \frac{R}{R_T}V_R S_1 - \frac{R}{R_T}(V_M sin(\theta_T - \theta_M) + U_S sin\theta_T - V_S cos\theta_T) - V_T C_0 + \frac{1}{2}V_T C_2 + \frac{1}{2}V_R S_2$$

(12)

$$A_n(n \geq 2) = \frac{R}{R_T}V_R C_n + \frac{1}{2}(V_T S_{n-1} + V_R C_{n-1} - V_T S_{n+1} + V_R C_{n+1})$$

(13)

$$B_n(n \geq 2) = \frac{R}{R_T}V_R S_n + \frac{1}{2}(-V_T C_{n-1} + V_R S_{n-1} + V_T C_{n+1} + V_R S_{n+1}) \tag{14}$$

The Fourier coefficients can be rearranged to obtain each wind component of the vortex:

$$V_R C_0 = \frac{A_0 + A_1 + A_2 + A_3 + A_4}{1 + \frac{R}{R_T}} - V_M cos(\theta_T - \theta_M) - V_R C_1 - V_R C_2 - V_R C_3 - \frac{R}{R_T}(U_S cos\theta_T + V_S sin\theta_T) \tag{15}$$

$$V_M cos(\theta_T - \theta_M) = A_0 - \frac{R}{R_T}V_R C_0 + \frac{1}{2}V_T S_1 - \frac{1}{2}V_R C_1 - U_S cos\theta_T - V_S sin\theta_T \tag{16}$$

$$V_T C_0 = -B_1 - B_3 + \frac{R}{R_T}[-V_M sin(\theta_T - \theta_M) + V_R S_1 + V_R S_3 - U_S sin\theta_T + V_S cos\theta_T] + V_R S_2 \tag{17}$$

$$V_T S_n = 2A_{n+1} - V_R C_n + V_T S_{n+2} - V_R C_{n+2} - 2\frac{R}{R_T}V_R C_{n+1} \tag{18}$$

$$V_T C_n = -2B_{n+1} + V_R S_n + V_T C_{n+2} + V_R S_{n+2} + 2\frac{R}{R_T}V_R S_{n+1} \tag{19}$$

Plugging Eq. 16 into Eq. 15:

$$V_R C_0 = \frac{A_0 + A_1 + A_2 + A_3 + A_4}{(1 - \frac{R^2}{R_T^2})} - \frac{A_0 + A_2 + A_4}{(1 - \frac{R}{R_T})}$$
$$- V_R C_2 - \frac{\frac{R}{R_T}}{1 - \frac{R}{R_T}}V_R C_4 - \frac{1}{2}(\frac{1}{1 - \frac{R}{R_T}})(V_T S_5 - V_R C_5) \tag{20}$$

Equations 15- 19 correspond to equations (16)-(20) in Jou et al. (2008) with the additional terms of storm motion on Eqs. 15 - 17. Equation 20 is an updated version of Eq. 15 to minimize the unknown terms after plugging in the $V_M cos(\theta_T - \theta_M)$. One caveat of the $V_R C_0$ updated form is that the axisymmetric radial wind cannot be derived when $R = R_T$ because of the singular point. The derivation shows that the storm motion is aliased on the components of mean wind, wavenumber 0 tangential and wavenumber 0 radial wind. Since the storm motion can be accurately estimated over successive radar volumes, the above equations can yield more accurate estimation of these wind components. However, the separation of storm motion and mean wind does not solve the underdetermined problem that the number of unknown variables is greater than the number of equations. We apply the same closure assumption in Lee et al. (1999) and Jou et al. (2008) that the asymmetric component of radial wind is much smaller than the asymmetric component of tangential wind, so the terms associated with radial wind

asymmetries can be ignored. This closure assumption may not be applicable within the boundary layer or outflow layer where the radial wind asymmetries can be substantial. Future research is required to improve this closure assumption and retrieve the asymmetric radial wind.

With the above equations, the along-beam component of the mean flow, axisymmetric (n=0) tangential wind, axisymmetric radial wind, and asymmetric tangential winds (n=1-2) can be retrieved by performing linear least squares fit on $V_d D/R_T$ in a TC centered cylindrical coordinate. All data within 1-km radius-wide annulus are included in the linear least squares fit. To deal with missing data in observational radar data and reduce the influence of outliers (Matejka and Srivastava, 1991), the truncation of the Fourier series follows Lee et al. (2000) (Table 2), which is consistent with the restriction of maximum allowable gap size in Lorsolo and Aksoy (2012). Lorsolo et al. have shown that the maximum allowable gap size varies with number of gaps and noise. If more gaps are present in the signal, the maximum allowable gap size is greater than originally suggested in Lee et al. (2000). We allowed the maximum wavenumber up to wavenumber 2 in the retrieved tangential wind in this study to reduce retrieval errors.

## 3.2 GVTD-simplex center finding

The GVTD algorithm can be highly sensitive to the center location. Jou et al. (2008) showed that the uncertainty of the center location cannot exceed 5 km, or about 20% of the radius of the maximum wind (RMW), in order to have a reasonable wind retrieval. There are several ways to identify TC center, such as the geometric center (Griffin et al., 1992), wind center (Wood and Brown, 1992), dynamic center (Willoughby and Chelmow, 1982) and vorticity center (Marks et al., 1992). The centers identified by different methods are not necessarily collocated, and the range of uncertainties is a few kilometers or more. Since both vorticity centers estimated from the GVTD technique and dynamic centers derived from the aircraft reconnaissance were available, the comparison of the different centers is required in order to have a better result of wind retrieval.

The GBVTD-simplex algorithm is a method to identify TC vorticity centers using single-Doppler radar data developed by Lee and Marks (2000). The simplex center is found by maximizing the mean tangential wind within an axisymmetric TC with three operations on a simplex: reflection, contraction, and expansion (Lee and Marks, 2000; Harasti et al., 2004). The GBVTD-simplex algorithm reduces the uncertainties in estimating TC position and improves the quality of the GBVTD-retrieved TC circulation. The deviation of the true centers to the GBVTD-simplex center in an idealized TC is approximately 340 m. In this study, since the GVTD technique has better wind field estimation, we conducted the GVTD-simplex method to estimate the centers following the GBVTD-simplex algorithm. By maximizing GVTD-retrieved mean tangential wind, using the GVTD technique to estimate the TC vorticity center has higher accuracy than the GBVTD technique.

The GVTD-simplex method is performed as the following procedure:

(1) Doppler velocities on a CAPPI were interpolated onto a cylindrical grid with 1-km radial spacing centered at a given TC center.

(2) Find the TC center possessing the highest mean tangential wind within an axisymmetric TC.

(3) Use three operations on a simplex : reflection, contraction, and expansion - to search for a new maximum or minimum in

the field around the simplex. We put the dynamic center from HRD as the first guess. The operation process would start from this point and find the circulation center.

To compare the performance of GVTD-simplex and GBVTD-simplex centers, the dynamic centers from HRD (Willoughby and Chelmow, 1982) are treated as the reference center because they are consistently centered at the geometric center of the storm over our analysis period compared to the simplex centers. The centers are interpolated from a few dynamic centers with a series of spline curves every two minutes, so the centers are connected into a continuous track. Figure 2b shows that the centers retrieved by the GVTD-simplex method are not consistent with that of the retrievals from the GBVTD-simplex method, which is due to the more accurate estimation of axisymmetric tangential wind. Although the GVTD-simplex centers follow more closely to the dynamic centers over the 35 hours of observation, the location of the GVTD-simplex center is still variable, similar as GBVTD-simplex center (Lee and Marks, 2000) and not fully consistent with different radar retrievals (such as KAMX vs. KMLB, and KMLB vs. KJAX). Since the dynamic centers are qualitatively and quantitatively better than the simplex centers in terms of consistency, and are independent of assumptions in either the airborne dual-Doppler and single-Doppler retrievals, our study utilized the dynamic centers to perform the GVTD technique and cylindrical decomposition of the airborne wind field.

## 4    Wind retrievals comparison between single Doppler and airborne dual Doppler analyses

### 4.1    Wavenumber 0 tangential wind retrieval

The dual Doppler analyses from the four aircraft passes into Hurricane Matthew are optimal to evaluate the performance of the GVTD technique because we can obtain all coefficients (Eqs. 16 to 20) from the Fourier decomposition of $V_T$ and $V_R$, known storm motion, and mean wind components, which ensures the comparability of the wind field. To evaluate the improved algorithm, the wind fields from the dual Doppler analyses were resampled into Doppler velocity that would be observed by the KAMX radar. The subsequent analyses use the observations at the altitude of 4 km, so the ground-based $0.5°$ radar elevation beam can detect the TC inner core. The wind fields were then retrieved from the resampled velocities using the original and improved GVTD algorithms (Table 1). Figure 3a and b show the Doppler velocity observed by the KAMX radar and the Doppler velocity projected from the dual Doppler analysis using the 1855 - 1940 UTC aircraft data as an example. Figure 3c and d show the reflectivity field from the KAMX radar observation and dual-Doppler analysis respectively. The single Doppler observations have much more missing data in the eye and the moat due to the reduced sensitivity to weak echoes at longer range from the radar. Additionally, the maximum Doppler velocity of single Doppler observations is not collocated with the maximum Doppler velocity of the resampled dual Doppler analysis. This discrepancy is believed to be due to temporal aliasing from the extended sampling period of the aircraft pass compared to the ground-based scanning. The issue of temporal aliasing of airborne Doppler analysis will be discussed in more detail in section 4.2. Despite the aliasing, the dual Doppler analysis provides a relatively complete, consistent wind field and can reasonably be assumed as the "truth" for the purposes of algorithm evaluation.

Figure 4 displays the results from the improved GVTD algorithm compared to the original GVTD algorithm and the "true" wavenumber 0 (axisymmetric) tangential wind derived from the four dual-Doppler analyses. The green dashed line (hereafter optimal solution) shows the improved GVTD retrieved axisymmetric tangential wind ($V_T C_0$) when all the terms are known. This optimal retrieval from the resampled $V_d D / R_T$ from the dual Doppler analysis using Eq. 17 has the least deviation from the "truth" (black line) in general. When the storm motion is removed from the $V_T C_0$ retrieval (blue dashed line, hereafter optimal but without storm motion) the axisymmetric wind in the inner eyewall (between 10 and 30 km) does not differ significantly from the "truth". As the radius increases, the storm motion term becomes more important to the retrieval of wavenumber 0 tangential wind. Between the radius of 55 to 65 km the impact of neglecting the storm motion term can be up to $3\text{-}4\,\mathrm{m\,s^{-1}}$. In the optimal retrieval, all terms in Eq. 17 are known, but in practice, the retrieved Fourier coefficients are underdetermined and a closure assumption is required to retrieve $V_T C_0$, typically by neglecting the cross-beam mean wind and asymmetric radial wind. The orange line (hereafter original GVTD) in Fig. 4 shows the result of the original GVTD algorithm invoking this closure assumption where only the $B_1$ and $B_3$ coefficients are used to retrieve $V_T C_0$. The red line (hereafter improved GVTD) shows the improvement to the GVTD algorithm by adding in the storm motion terms, while still invoking the necessary closure assumption. The deviations from the "truth" caused by the closure assumption occur at different radii depending on the aircraft pass, suggesting that the neglected terms have different magnitudes in different parts of the storm. For example, in pass 1 the deviations are most pronounced from 55 - 70 km radius, while in pass 4 the deviations are most pronounced from 15 - 25 km radius.

Root-mean-square (RMS) differences of the above $V_T C_0$ solutions and integrated perturbation pressure deficit (optimal, optimal but without storm motion, original GVTD and improved GVTD algorithms) averaged over the four passes are shown in Table 3. For RMS difference of $V_T C_0$, the optimal solution has the least deviation from the "truth". Including the storm motion terms decreases the RMS differences about $0.8\,\mathrm{m\,s^{-1}}$ in both optimal versus optimal without storm motion and improved GVTD versus original GVTD. Interestingly, RMS difference of improved GVTD algorithm has a similar magnitude as the optimal solution but without storm motion, suggesting that the influence from storm motion herein is similar as neglecting terms in Eq. 17.

The perturbation pressure deficit is integrated from r = 10 to 70 km using the gradient wind balance equation (Lee et al., 2000). The integrated perturbation pressure deficit retrievals from different methods agree well with the "truth" ($\sim$ 1 mb RMS in general). Similar as the results of RMS difference of $V_T C_0$, including the storm motion terms decreases the RMS differences about 0.2 mb. The RMS difference of improved GVTD algorithm from the single Doppler retrieval has the least deviation from the "truth", suggesting that the perturbation pressure deficit derived from the single Doppler observations has high fidelity.

The $V_T C_0$ retrieved by the original (orange dots) and improved (red dots) GVTD algorithm from the KAMX observations alone (Fig. 3a) are also shown in Fig. 4. The axisymmetric tangential winds retrieved from the KAMX radar alone are in generally good agreement with the winds from the dual Doppler analyses. As in the dual Doppler retrieval, the storm motion term has relatively small impact on the retrieval in the inner eyewall because $R/R_T$ is small. The deviation between the retrievals becomes larger beyond the radius of 40 km, but the improved GVTD algorithm is generally more consistent with

the "truth" compared to the original GVTD algorithm. RMS differences show a reduction $0.4\,\mathrm{m\,s^{-1}}$ between the original and improved GVTD algorithms compared to the "truth" (Table 3).

A nonparametric Wilcoxon signed-rank test is conducted to test the null hypothesis that two paired sets of the RMS differences derived from the original and improved GVTD algorithms are drawn from the same distribution. The RMS difference between the original and improved GVTD algorithms is statistically significant with a $p$ value $< 0.001$ using both the projected dual-Doppler winds and single-Doppler velocities, indicating that we can reject the null hypothesis at the $1\%$ significance level ($99\%$ confidence). The statistics suggest that the RMS differences distribution of wavenumber 0 tangential wind retrieved from the original GVTD algorithm are likely to be larger than those from the improved GVTD method. While it is a relatively small reduction in the RMS difference in the current case, the statistically significant difference in this algorithm error contributes to an overall reduction in the total error from instrument, algorithm, and sampling contributions. The comparison demonstrates that the inclusion of the storm motion has better consistency of the wavenumber 0 tangential wind intensity with the dual-Doppler reference solution, and that the improved GVTD technique can provide insightful information about TC kinematic structure from ground-based single Doppler radar data.

### 4.2 Asymmetric wind retrievals

Both Lee et al. (1999) and Jou et al. (2008) have tested the GBVTD and GVTD techniques respectively with analytic datasets and confirmed the accuracy of these two methods. Murillo et al. (2011) further assessed the ability of GBVTD to retrieve low-wavenumber wind structure by comparing simultaneous single Doppler and dual-Doppler retrievals in Hurricane Danny (1999) using ground-based radar observations. However, no previous studies to the authors' knowledge have conducted a detailed comparison of the Fourier decomposition of the wind field between the ground-based single Doppler and airborne dual-Doppler wind retrievals. We have already shown that wavenumber 0 tangential wind from the dual Doppler analyses can be accurately retrieved by the GVTD-improved algorithm in section 4.1. With the confidence of the performance of the GVTD technique, the next step is to examine the low-wavenumber structure and applicability of a steady-state assumption of the dual Doppler analysis.

Figure 5a displays $V_dD/R_T$ on a ring at the RMW (18 km) derived from the dual Doppler and single Doppler analyses. For clarity, we use the term "harmonics" to refer to Fourier decomposition in $V_dD/R_T$ around the ring, and "wavenumbers" to refer to the components of tangential wind which are constructed from combinations of the observed Doppler harmonics. $V_dD/R_T$ constructed from both analyses are dominated by a single harmonic pattern with very similar magnitude. Nevertheless, a small difference of $V_dD/R_T$ pattern between the two analyses is evident. The discrepancies of the pattern and magnitude suggest that higher harmonics of the $V_dD/R_T$ coefficients are different.

Figure 5b shows the residuals after the subtraction of harmonics 0 and 1 components from the total $V_dD/R_T$, with the harmonic 2 components of $V_dD/R_T$ highlighted. The residuals include the harmonic 2 and higher components of $V_dD/R_T$. The residuals from the single Doppler analysis (light red line) are dominated by a wavenumber 2 component, while the residuals from the dual Doppler analysis (light green line) are without a clear pattern. The peak value of the residual from the single Doppler analysis is $8\,\mathrm{m\,s^{-1}}$, but the peak value of the residual from the dual Doppler analysis is about $3\,\mathrm{m\,s^{-1}}$. To quantitatively

compare the two retrievals, Table 4 shows the retrieved magnitude of harmonics 0 - 3 around the RMW. The magnitude of harmonics 0, 1, and 3 are similar at the RMW for the two analyses with around $1\,\mathrm{m\,s^{-1}}$ deviation. The similarity in harmonic 1 is primarily responsible for the similarity in the wavenumber 0 tangential wind (Fig. 4). Harmonic 2 shows a much larger deviation of $6\,\mathrm{m\,s^{-1}}$, resulting in different wavenumber 1 tangential wind retrievals.

Figure 6 shows the harmonics of $V_d D/R_T$ as a function of radius for the resampled dual Doppler wind field (green dashed line) and single Doppler observations (red dots). Note that the ordinate on each panel is different due to the varying magnitudes of each coefficient. The deviations of $A_0$, $A_1$ and $B_1$ coefficients between the two analyses within the eyewall region (15 - 25 km) are less than $2\,\mathrm{m\,s^{-1}}$. The deviations in $A_0$, $A_1$ and $B_1$ coefficients are larger from the eye to the inner edge of the eyewall (10 - 15 km) and outside of the eyewall region (beyond 25 km), but generally have a similar magnitude. On the other hand, larger discrepancies are apparent with $A_2$ and $B_2$. $A_2$ is similar between 10 to 20 km, but has a large $7\,\mathrm{m\,s^{-1}}$ difference outside 20 km. $B_2$ is different at nearly all radii, with up to $7\,\mathrm{m\,s^{-1}}$ differences. The different amplitudes of $A_2$ and $B_2$ indicate that the retrieved wavenumber 1 tangential wind phase and magnitude are inconsistent (Eq. 19). Discrepancies are also evident in the $A_3$ and $B_3$ coefficients, indicating inconsistencies in the retrieved wavenumber 2 tangential wind phase and magnitude as well.

We hypothesize that the discrepancies of retrieved wavenumber 1 and 2 tangential winds are due to the steady state assumption in the dual Doppler wind synthesis into one snapshot. Two different types of sampling errors in effect arise from the steady state assumption – the first is due to the time lag between fore and aft beams, and the second is due to the length of time used to composite the multi-Doppler into a single snapshot. While both produce some errors, the latter is more consequential when considering the temporal evolution of the phenomena is faster than the period of data collection, resulting in evolution over the flight pass. The "local" wind may be correct, but the overall structure is distorted by collapsing to a single time. For example, the propagation velocity of a wavenumber 2 vortex Rossby wave (VRW) is half of the symmetric tangential wind velocity (Lamb, 1932; Guinn and Schubert, 1993; Kuo et al., 1999). A propagating wavenumber 2 asymmetry could then alias onto other wavenumbers, contributing to a discrepancy in wavenumber 1 tangential wind.

To test the hypothesis, the phases of maximum wavenumber 1 and wavenumber 2 tangential winds retrieved from the 5-minute single Doppler observations are examined in Fig. 7 for the temporal evolution during the first flight pass from 1907 to 1940 UTC. The amplitude and phase (Eqs. 18 and 19) of wavenumber 1 and 2 tangential winds are denoted in polar coordinates by the radius and azimuth, respectively. The wavenumber 1 tangential wind (Fig. 7a) generally stayed unchanged throughout the first pass with a magnitude between 8 and $12\,\mathrm{m\,s^{-1}}$ and phase to the E to NE (same as the wavenumber 1 reflectivity, not shown here). Environmental vertical wind shear derived from the Statistical Hurricane Intensity Prediction Scheme dataset (SHIPS) points to the northeast direction with a magnitude of $7\,\mathrm{m\,s^{-1}}$, suggesting that the wavenumber 1 distribution is forced by the vertical wind shear to be consistently in the downshear-right quadrant.

The wavenumber 2 tangential wind (Fig. 7b) propagated cyclonically during the flight pass (same as the wavenumber 2 reflectivity, not shown here) with a magnitude up to $7\,\mathrm{m\,s^{-1}}$. The propagation of the wavenumber 2 tangential wind is estimated to be 285 degrees from 1907 to 1940 UTC, which is $35\,\mathrm{m\,s^{-1}}$, or 63% of $V_{Tmax}$. The propagation of the wavenumber 2 tangential wind is roughly consistent with linear VRW theory (Kuo et al., 1999; Cha et al., 2020).

According to the single Doppler analysis, the wavenumber 2 VRW propagated approximately 43 degrees every five minutes on average. The wavenumber 2 maximum amplitudes were located in the NE and SW quadrants as the P3 approached the inner eyewall from the SW (Fig. 2), such that the TDR would see the maximum component in the SW eyewall. The VRW rotated cyclonically as the P3 crossed the 36 km diameter eye at $120\,\mathrm{m\,s^{-1}}$, such that by the time the P3 exited the eyewall the radar would begin to see the wavenumber 2 wind minimum on the NE side. The SW maximum and NE minimum would then appear as a wavenumber 1 component under a steady state assumption. The analysis supports our hypothesis that the propagation of wavenumber 2 tangential wind is aliased onto the steady wavenumber 1 component, resulting in a reduced amplitude and a phase shift in $A_2$ and $B_2$ in the dual Doppler analysis. Since the axisymmetric radial wind is influenced by the harmonics 2 and 3, and wavenumber 2 radial wind component (Eq. 20), we cannot fully validate the axisymmetric radial wind retrieval with the current dataset. Lee et al. (2006) shows that the Lamb solution of the wavenumber 2 radial wind has comparable magnitude as the wavenumber 2 tangential wind but with a phase shift, so the wavenumber 0 radial wind retrieval is uncertain when VRWs are present. The evaluation for the accuracy of the axisymmetric radial wind retrieval is not included in this study.

The above analysis suggests that both the dual Doppler and single Doppler analysis have strengths and weaknesses. The retrieved wavenumber 0 component of tangential wind has been shown to be comparable between an airborne dual Doppler wind synthesis and single Doppler retrieval due to the fact that a steady state assumption for one flight pass is usually valid to apply to the axisymmetric circulation. The axisymmetric kinematic structure of Hurricane Matthew revealed in the dual Doppler analysis is consistent with the single Doppler retrievals across multiple aircraft passes (Cha, 2018). However, the steady-state assumption may not be applicable with more rapidly evolving features, such as convective bursts, rotation of mesovortices, and propagation of a wavenumber 2 or higher VRWs. The evolution during the aircraft pass could potentially impact the asymmetric reflectivity and wind structure retrieval from a dual or multi-Doppler analysis due to temporal aliasing. On the other hand, dual Doppler analysis can retrieve high-quality and detailed three-dimensional structure, whereas a ground-based single Doppler radar retrieval is often limited by spatial aliasing and the observing distance from the radar to TC. Nevertheless, the analyses presented herein demonstrate that complementary information can be retrieved from both single and dual Doppler retrievals.

### 4.3 An idealized experiment with a propagating wavenumber 2 asymmetry

To further test our hypothesis, we performed an idealized experiment to sample a rotating wavenumber 2 asymmetry by an aircraft with a realistic scanning strategy, similar to that first presented by Bell et al. (2007). The idealized framework allows for minimization of instrument and algorithm errors in order to isolate the sampling error due to the steady-state assumption. The simulated aircraft penetrates the TC "eyewall" at 120 m s$^{-1}$ airspeed on a south to north flight track, which is a typical straight-line flight track during NOAA P3 operational reconnaissance. The radar has "perfect" FAST scanning with no instrument error and ideal geometry from the straight flight track, such that a steady-state wind field can be retrieved with minimal error (not shown). The TC wind field is a symmetric Rankine vortex with a 20 km RMW and $V_{max}$ of $50\,\mathrm{m\,s^{-1}}$, representative of a strong mature TC, with an added propagating wavenumber 2 linear Rankine edge wave with an epsilon of 3 km corresponding to an amplitude of 7.1 m s$^{-1}$ as in Lee et al. (2006). The initial phase of the asymmetry is oriented from east to west, and it rotates

around the RMW cyclonically at 1/2 $V_{max}$. The simulated aircraft takes 22 minutes to cross the 160 km distance through the storm, during which time the wavenumber 2 asymmetry rotates to have a north-south orientation. The time-mean phase of the asymmetry is therefore 45 degrees, oriented from southwest to northeast over the course of the sampling period.

Figure 8a shows the derived tangential wind for a wavenumber 2 rotation during the flight pass. The overall magnitude of the wind field is close to the prescribed time-mean structure but is distorted, suggesting temporal aliasing from the extended sampling period of the aircraft pass due to the propagation of the wavenumber 2 asymmetry. Figure 8b - d shows the retrieved wavenumber 1, 2 and 3 tangential wind fields from the Fourier decomposition. Although only a wavenumber 2 component is prescribed in the experiment, the retrieved wavenumber 1 and 3 components are up to 1.5 - 2 m s$^{-1}$. The orientation of the flow is close to the "true" time-mean structure shown in black contours, and the Fourier wavenumber 2 component has the generally correct phase. We note that the amplitude of the wavenumber 2 component correctly changes sign at the RMW due to the construction of a pure vorticity wave and is consistent with the prescribed flow and phase. The exact asymmetry retrieved depends to a large extent on the specifics of the flight track. However, we note that over the 22 minute period, a ground-based single Doppler retrieval would see the storm ∼4 times with different phase orientations of the asymmetry (not shown). While idealized, this experiment provides further evidence that the steady-state assumption to synthesize the data into one snapshot with the presence of evolving features could result in temporal aliasing and result in potential misinterpretation of the asymmetric flow field.

## 5    Conclusions

Tropical cyclone (TC) wind retrieval techniques from Doppler radar observations are salient to identify the convective and kinematic structure and evolution of TCs. The current study is believed to be the first detailed comparison of the retrieved wind field between ground-based single Doppler and airborne dual-Doppler wind techniques. Hurricane Matthew (2016) was observed by the polarimetric Next Generation Weather Radar (NEXRAD) in Miami (KAMX) and NOAA P-3 airborne radar near the coast of Florida for several hours, providing a novel opportunity to evaluate and compare single and dual Doppler wind retrieval techniques for TC flows.

Jou et al. (2008) has shown that the GVTD technique improves the capability of the GBVTD and can retrieve a subset of the wind field from a single ground-based Doppler radar under the assumption of nearly axisymmetric rotational wind. Here we present an improved technique that mitigates errors due to storm motion, which yields a more accurate estimation of mean wind, wavenumber 0 tangential and radial winds. The aliasing of unresolved wind components from the single Doppler measurements remains, so another closure assumption for the GVTD technique is needed to improve the accuracy of the retrieval in the future. In addition to the wind improvements, the GVTD-simplex centers follow more closely to the dynamic centers compared to GBVTD-simplex centers due to the more accurate estimation of axisymmetric tangential wind. However, the GVTD-simplex centers are still variable and not fully consistent between retrievals from different radars (such as KAMX vs. KMLB, and KMLB vs. KJAX), suggesting further improvement is needed in the technique. Nevertheless, the improvement

of TC wind estimation from single Doppler radar presented herein can be useful in deducing storm structure for research purposes and assimilating real-time TC intensity and structure for forecasts (Zhao et al., 2012).

The strengths and weaknesses of different platform observing capabilities and retrieval techniques for TC wind fields are discussed in this study. The full three-dimensional wind field from airborne radar fore-aft pseudo-dual Doppler scanning can be retrieved by a spline-based technique called SAMURAI, but is shown to have errors due to temporal aliasing from the non-

455 simultaneous Doppler measurements when the temporal evolution of the phenomena is faster than the period of data collection. Single Doppler wind retrieval by the GVTD technique can be obtained at a 5-minute interval, but spatial aliasing remains due to the limited measurements from only radial direction of the radar beam. A comparison between the two techniques shows that the retrieved axisymmetric (wavenumber 0) component of tangential winds are generally comparable between the two techniques, and the improved GVTD technique improves the accuracy of the retrieval. Fourier decomposition of asymmetric

kinematic and convective structure shows more discrepancies between the two techniques due to spatial and temporal aliasing in the retrievals. The propagation of wavenumber 2 tangential wind is found to be aliased onto the steady wavenumber 1 component retrieved by the dual Doppler analysis, causing a reduced amplitude and phase shift of wavenumber 1 tangential wind. The steady-state assumption for low-wavenumber structure retrievals from pseudo-dual Doppler analysis may not be applicable with the presence of rapidly evolving features, and the temporal evolution during a flight pass should be considered.

On the other hand, the steady-state assumption is less severe for single Doppler wind retrievals, and the rapidly evolving phenomena, such as vortex Rossby waves, have been documented by several studies using the single Doppler observations (Corbosiero et al., 2006; Cha et al., 2020). The analyses presented herein demonstrate that complementary information can be retrieved from both single and dual Doppler retrievals. Future work on the time-dependent analysis of asymmetric structure in dual Doppler retrieval and asymmetric flow closure assumptions in single Doppler retrieval are required to reconcile differences

in retrieved TC structure.

*Code and data availability.* The underlying Julia code used for analysis in this study is available from the author upon request. The Vortex Objective Radar Tracking And Circulation (VORTRAC) and SAMURAI software are part of the Lidar Radar Open Software Environment (LROSE) and are available on the web: http://lrose.net. The data used in this paper are available through zenodo: https://doi.org/10.5281/zenodo.4427194.

*Author contributions.* The study was designed by both authors. TC carried out the analysis and made the figures supervised by MB. TC
wrote the manuscript with contributions from MB.

*Competing interests.* The authors declare that they have no conflict of interest.

*Acknowledgements.* This research was supported by National Science Foundation award OAC-1661663. We would like to thank NOAA Aircraft Operations Center and the Hurricane Research Division of the Atlantic Oceanographic and Meteorological Laboratory for collecting the airborne tail Doppler radar data used for this study, and the National Weather Service for the ground-based radar data. We thank Naufal
Razin and three anonymous reviewers for their constructive and helpful comments to improve the quality of the manuscript.

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

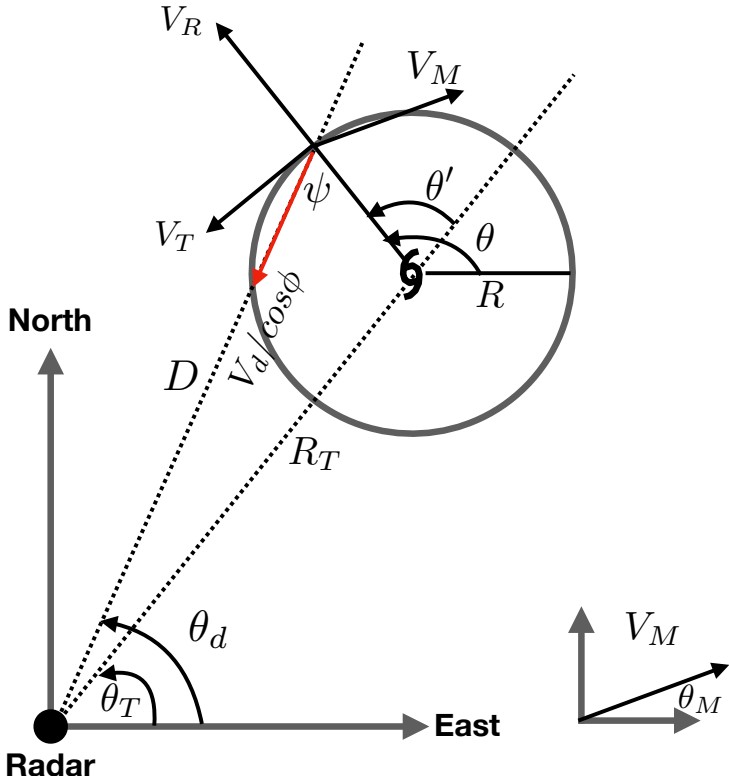

**Figure 1.** The geometry and symbols used in the formulation of GVTD (modified from Jou et al. (2008)). Red arrow denotes the Doppler velocity. Symbols are defined in the text.

**Table 1.** Details of aircraft missions and corresponded KAMX ground-based radar observation period for this study.

| Radar analysis | Duration |
| --- | --- |
| P3 | 1855-1940 UTC 6 Oct 2016 |
| P3 | 2020-2105 UTC 6 Oct 2016 |
| P3 | 2145-2230 UTC 6 Oct 2016 |
| P3 | 2305-2340 UTC 6 Oct 2016 |
| KAMX | 1907-0550 UTC 6-7 Oct 2016 |

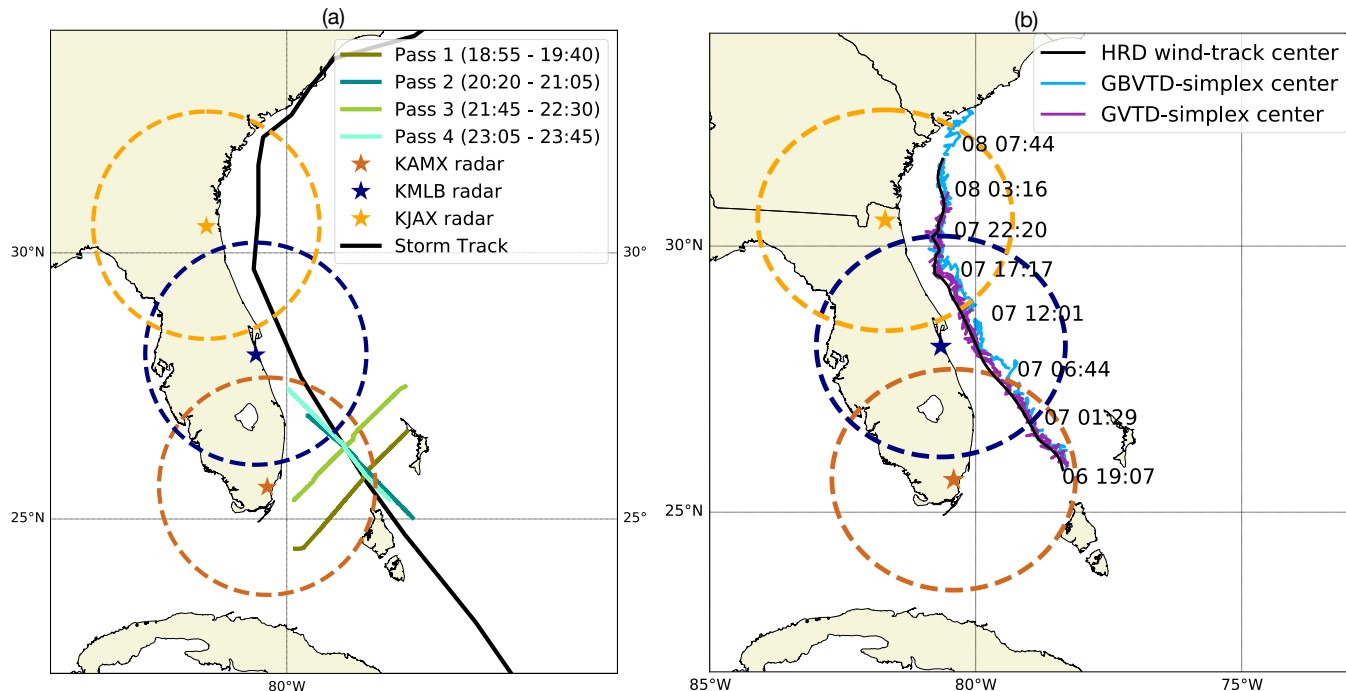

**Figure 2.** (a) Hurricane Matthew's center track from best track data (black line) and different radar and aircraft estimates. Passes 1-4 denote the consecutive flight segments of the P3 aircraft across the cyclone on 6 October. Colored circles and stars represent the ground-based radar detecting range (230 km) and location of single Doppler radar respectively. (b) Comparison of Hurricane Matthew's track between HRD dynamic aircraft center (black line), GBVTD-simplex center (light blue line), and GVTD-simplex center (purple line). Colored circles and stars are the same as (a).

**Table 2.** The maximum allowable data gap determines the maximum wavenumber used in the least squares fit.

| Wavenumber | Gap($^{o}$) |
|:----------:|:-----------:|
| 0 | $\leq 180$ |
| 1 | $\leq 90$ |
| 2 | $\leq 60$ |

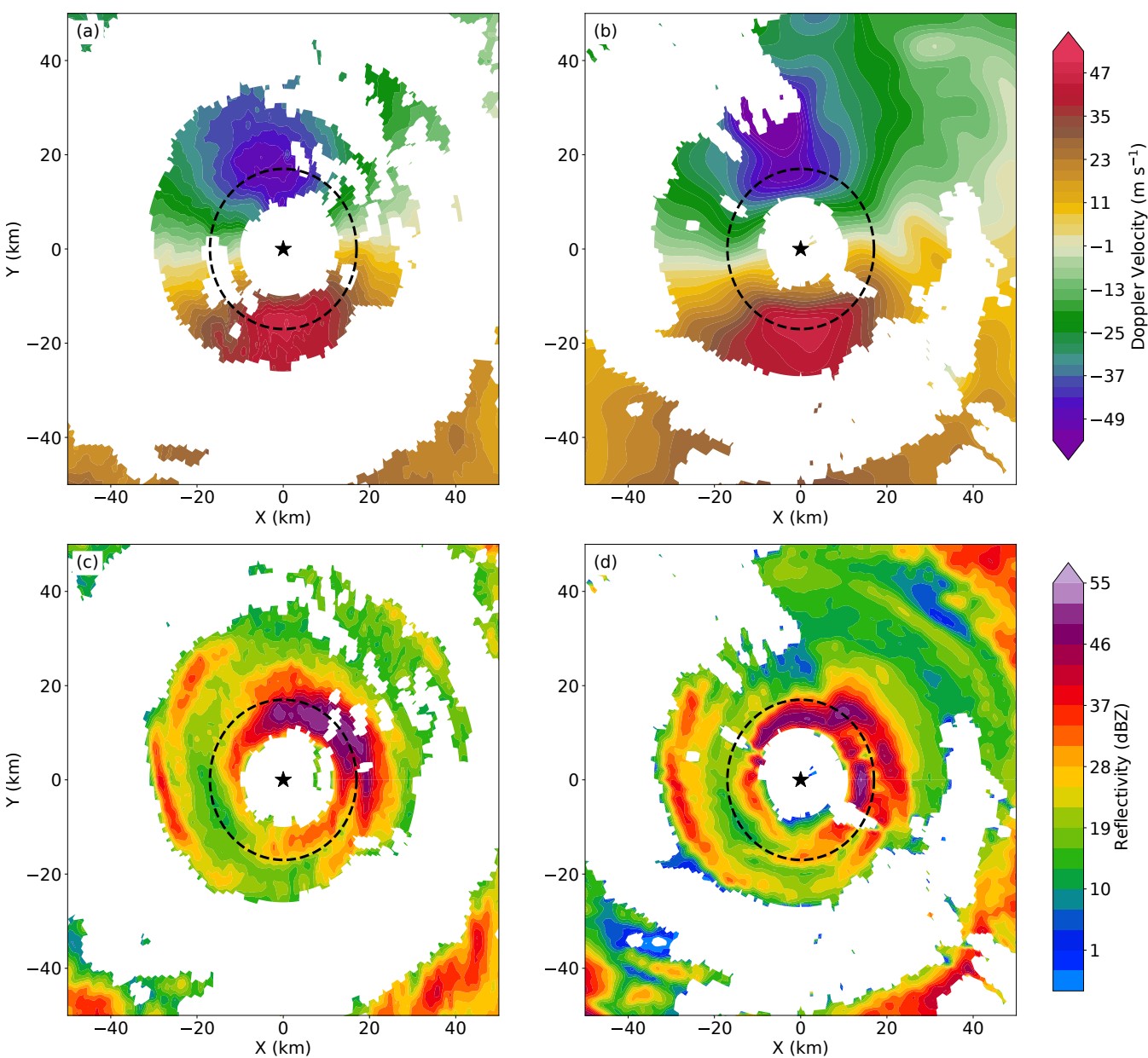

**Figure 3.** Doppler velocity at 4 km constant altitude (a) observed by the KAMX radar at 1921 UTC, and (b) resampled from the dual Doppler analysis synthesized from 1855 - 1940 UTC. Reflectivity at 4 km altitude (c) observed by the KAMX radar and (d) derived from the dual Doppler analysis. The timing of (c) and (d) are the same as (a) and (b), respectively. The black star denotes the TC center, and the dashed circle denotes the radius of maximum wind of 18 km.

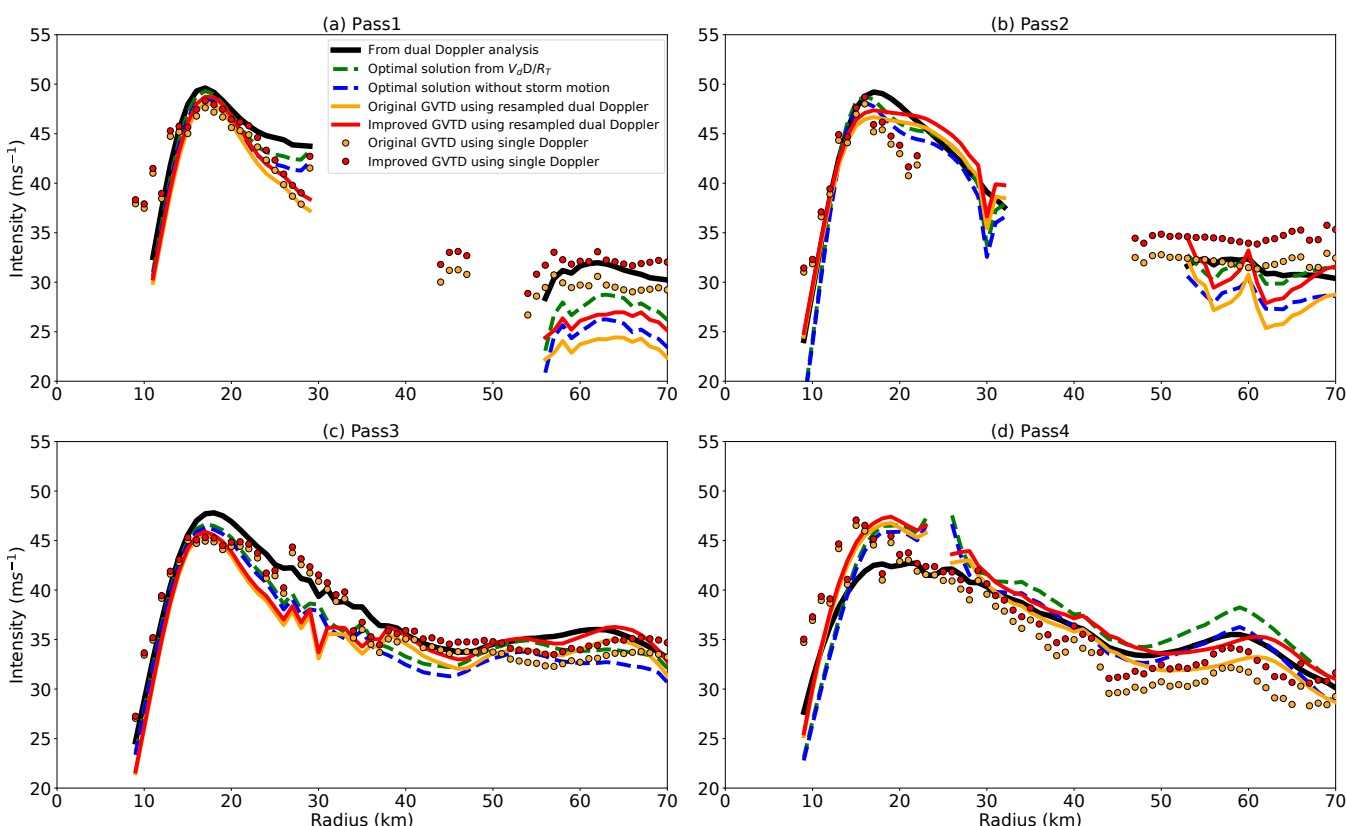

**Figure 4.** Wavenumber 0 tangential wind retrieved by various techniques at (a) Pass 1 from 1855 to 1940 UTC, (b) Pass 2 from 2020 to 2105 UTC, (c) Pass 3 from 2145 to 2230 UTC, (d) Pass 4 from 2305 to 2340 UTC. The retrievals are the Fourier decomposition of tangential wind from the dual Doppler analysis (black line), the Fourier decomposition of $V_d D/R_T$ (green dashed line), the Fourier decomposition of $V_d D/R_T$ without storm motion (blue dashed line), the original GVTD algorithm (orange line) and improved GVTD algorithm (red line) from the resampled dual Doppler analysis, and the original GVTD algorithm (orange dot) and improved GVTD algorithm (red dot) from the single Doppler observations.

**Table 3.** The averaged root mean squared error of the $V_T C_0$ and integrated perturbation pressure deficit retrieved from different methods described in Fig. 4 from the four flight passes and single Doppler radar observations. The errors are calculated from r = 10 - 70 km on z = 4 km.

| RMSE | P-3 dual-Doppler | | | | NEXRAD single Doppler | |
|---|---|---|---|---|---|---|
| | Optimal | Optimal but no storm motion | Origial GVTD | Improved GVTD | Origial GVTD | Improved GVTD |
| $V_T C_0$ (m s$^{-1}$) | 1.49 | 2.37 | 3.17 | 2.3 | 2.81 | 2.46 |
| Integrated perturbation pressure deficit (mb) | 0.91 | 1.14 | 1.3 | 1.07 | 0.84 | 0.68 |

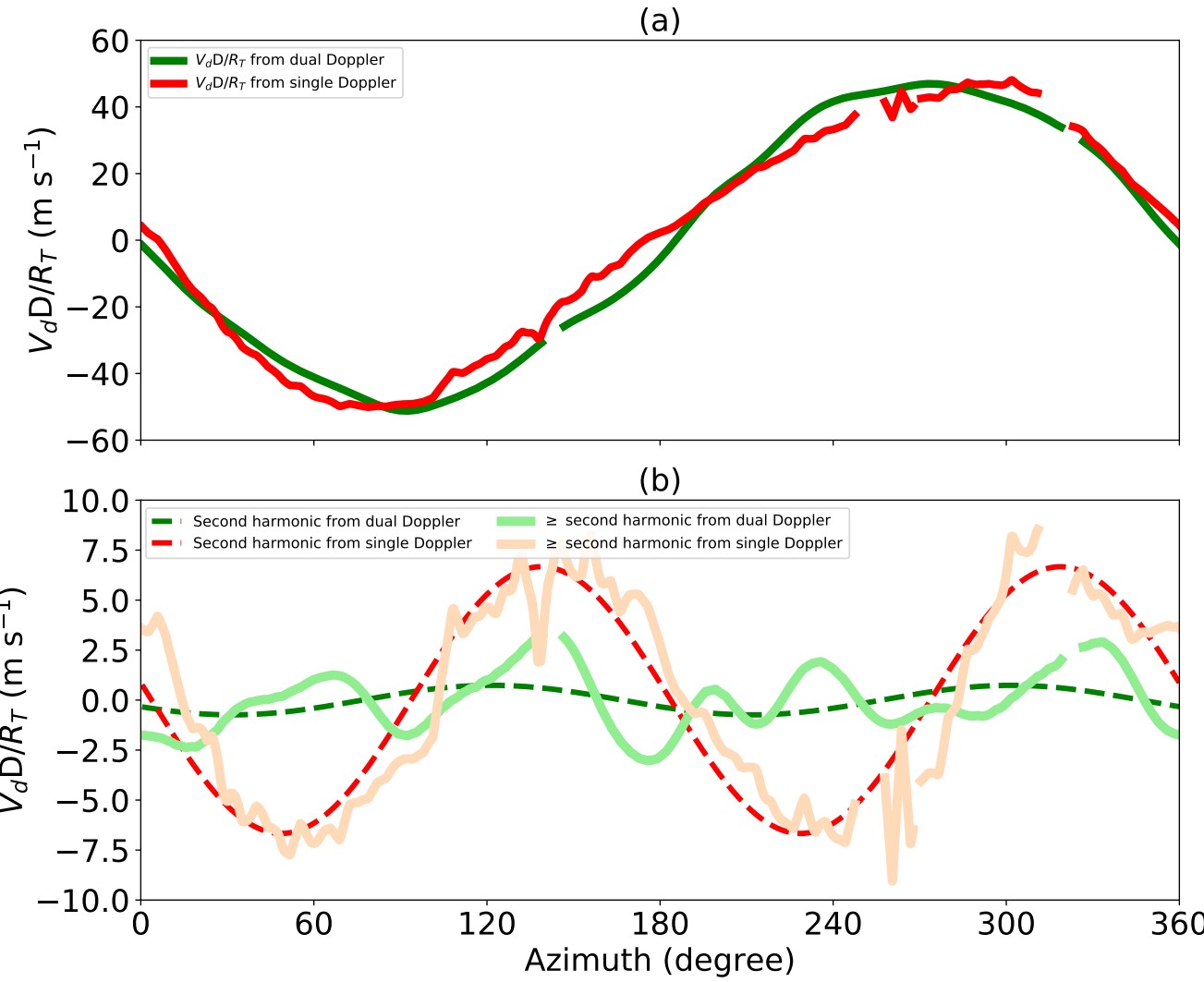

**Figure 5.** (a) Comparison of retrieved $V_dD/R_T$ between the dual Doppler and single Doppler analyses at the RMW of 18 km. Green line denotes retrieved $V_dD/R_T$ from the resampled dual Doppler analysis synthesized from 1855 - 1940 UTC, and red line denotes retrieved $V_dD/R_T$ from the KAMX radar at 1921 UTC. (b) Comparison of retrieved second harmonic and higher components of $V_dD/R_T$ between the dual Doppler and single Doppler analyses at the RMW. The solid line denotes the residuals of subtracting the harmonics 0 and 1 from $V_dD/R_T$. The dashed line represents the harmonic 2 of $V_dD/R_T$.

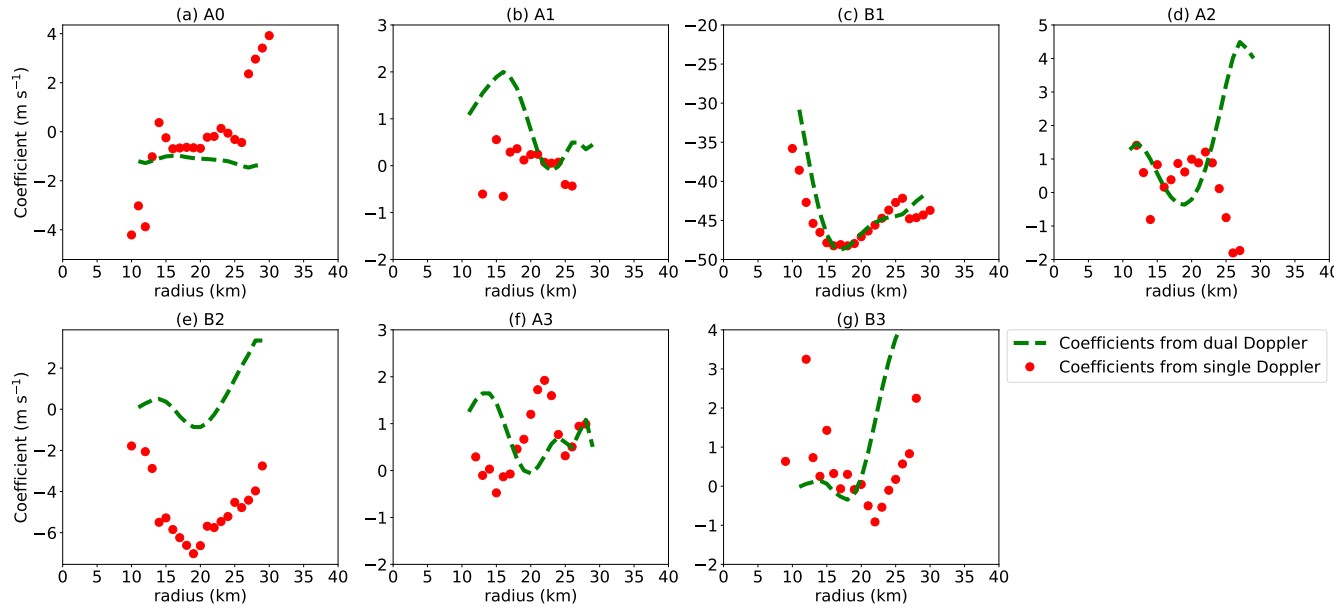

**Figure 6.** Comparison of $V_d D/R_T$ harmonics coefficients derived from the resampled dual Doppler winds synthesized from 1855 - 1940 UTC using the improved GVTD technique (green dashed line), and single Doppler retrieval at 1921 UTC using the improved GVTD technique (red dot). (a) $A_0$ (to obtain $V_R C_0$ and $V_M cos(\theta_T - \theta_M)$) (b) $A_1$ (to obtain $V_R C_0$) (c) $B_1$ (to obtain $V_T C_0$) (d) $A_2$ (to obtain $V_R C_0$ and $V_T S_1$) (e) $B_2$ (to obtain $V_T C_1$) (f) $A_3$ (to obtain $V_R C_0$ and $V_T S_2$) (g) $B_3$ (to obtain $V_T C_0$ and $V_T C_2$)

**Table 4.** $V_d D/R_T$ harmonics coefficients amplitude (harmonics 0 to 3) retrieved from the single Doppler and dual Doppler analyses.

| Coefficient magnitude (m s$^{-1}$) | Single Doppler retrieval | Dual Doppler retrieval |
|---|---|---|
| $A_0$ | 0.26 | -1.08 |
| $\sqrt{A_1{}^2 + B_1{}^2}$ | 46.29 | 47.61 |
| $\sqrt{A_2{}^2 + B_2{}^2}$ | 7.05 | 0.93 |
| $\sqrt{A_3{}^2 + B_3{}^2}$ | 0.67 | 0.19 |

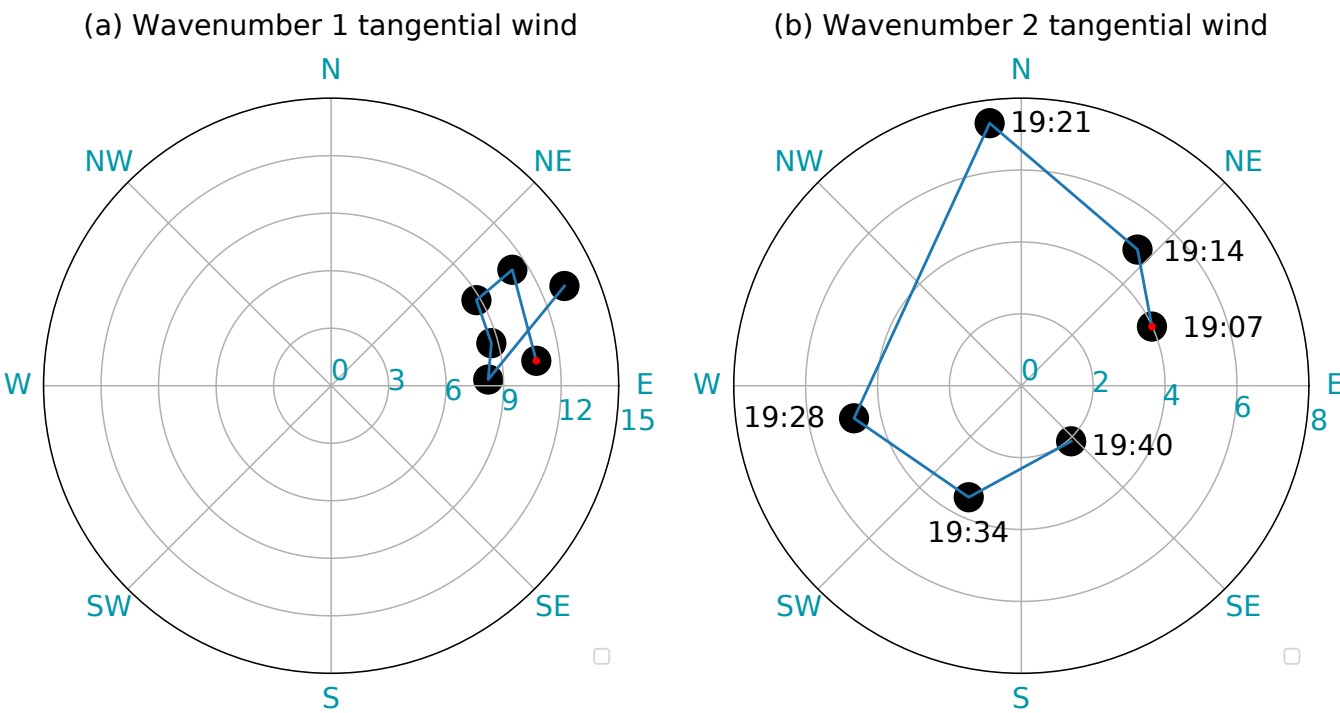

**Figure 7.** Azimuth and amplitude polar diagram of the temporal evolution of (a) maximum wavenumber 1 tangential wind amplitude and phase and (b) maximum wavenumber 2 tangential wind amplitude and phase in the inner eyewall (<18 km) derived from the GVTD single Doppler analysis from 1907 to 1940 UTC 6 October. The amplitude of the tangential wind component is denoted by the radius of each dot, with the phase denoted by the azimuth. The red dot in (a) and (b) indicates the starting time at 1907 UTC, and the temporal evolution follows the blue line.

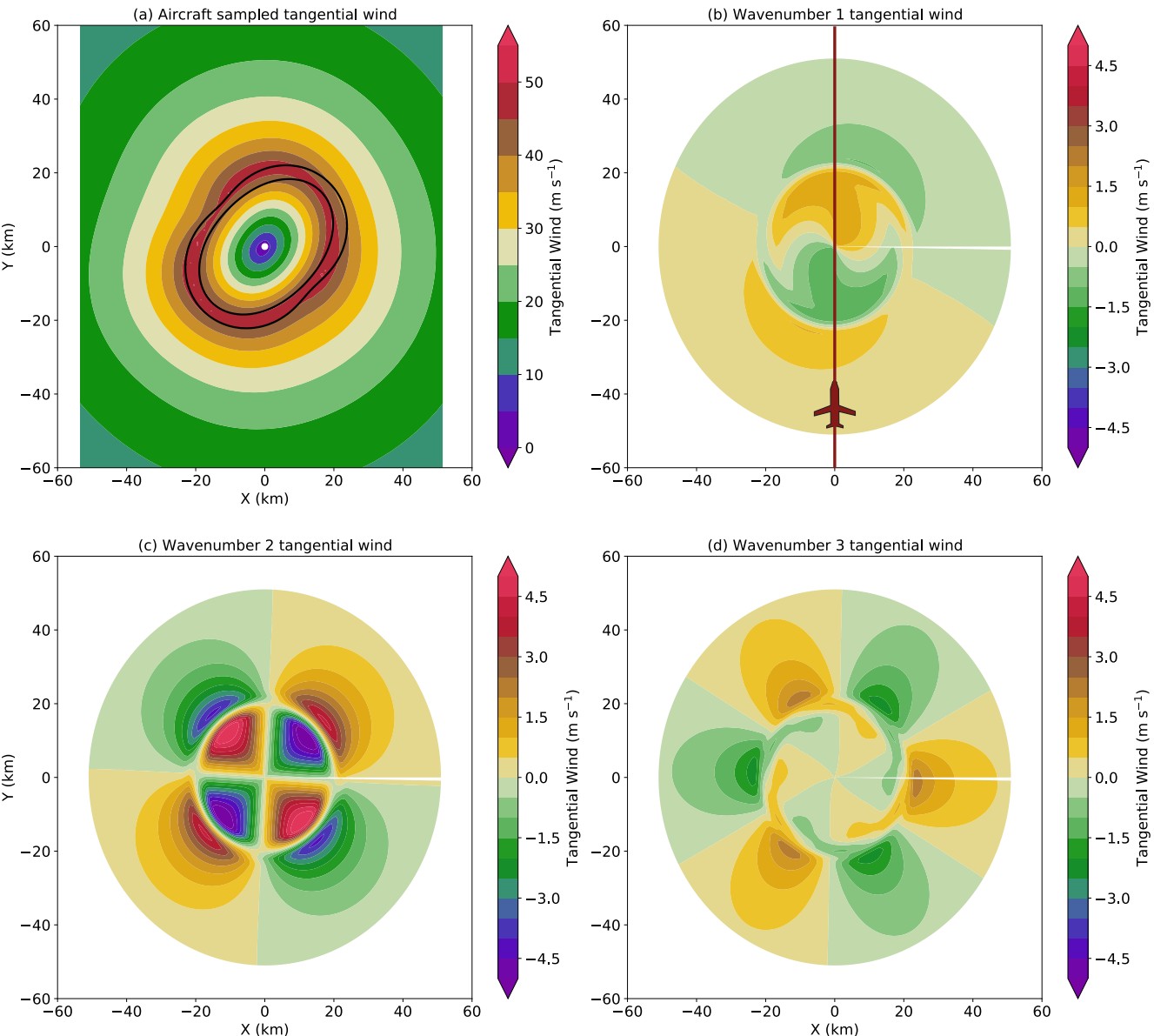

**Figure 8.** (a) Idealized dual-Doppler tangential wind speed retrieved from a straight flight pass through propagating wavenumber two asymmetry (color in m s$^{-1}$) and 45 m s$^{-1}$ contour of prescribed, time-averaged wavenumber two in black. The initial phase of the propagating wavenumber 2 tangential wind is oriented from east to west and final phase is from north to south, resulting in a time-averaged southwest to northeast orientation. (b) Wavenumber 1 plus aircraft flight track from south to north, (c) Wavenumber 2, and (d) Wavenumber 3 tangential wind components retrieved by the Fourier decomposition of the tangential wind field shown in panel (a).