# Peer review of "Comparison of Single Doppler and Multiple Doppler Wind Retrievals in Hurricane Matthew (2016)"

_Atmospheric Measurement Techniques, 2020_

## Referee Comment (RC1) · Anonymous Referee #1 · 19 Sep 2020

There are some interesting and useful components of this manuscript related to differences in ground-based and airborne wind retrieval methods. The writing and organization are clear, for the most part. However, there are several major concerns with the paper as listed below that will take a significant amount of time to address. I therefore recommend that the paper be declined and re-submitted once these issues are addressed.

Major:

(1) The total error in the radar comparisons is a summation of instrument effects (e.g., signal-to-noise ratio), sampling effects (e.g., gaps in data coverage, spatial/temporal resolution) and algorithm effects (e.g., geometry, approximations, solution method). The comparisons between dual Doppler and single Doppler retrievals for WV#0 tangential winds have some differences in the eyewall region (especially pass 4) that appear to be due to differences in the GVTD method (algorithm effect). The authors are trying to isolate the effects of the "steady-state" assumption (sampling effect) in airborne radar analysis by comparing WV#1 and WV#2 tangential winds from dual Doppler and single Doppler retrievals.  However, it is not at all clear how much of the differences the authors are seeing are due to instrument effects, algorithm effects or other sampling effects. The WV#1 and WV#2 tangential winds should have larger errors due to algorithm effects when compared to the WV#0 tangential winds. This makes it very difficult or impossible to isolate the effects of just the steady-state assumption and thus the conclusions from this analysis are uncertain. The authors need to isolate these effects through some type of simulated analysis in order to make definitive conclusions.

(2) Regarding the improved GVTD method: the authors have done a thorough analysis of the impacts of storm motion on the ground-based retrieval. I think this analysis is useful for the community. However, I think the revised method only makes minor improvements on the errors in the retrieved mean tangential winds, but the authors have overstated their importance at several places in the paper, including the abstract.  Thus, the tone of the paper needs to be revised in several places and more details are found in the additional comments section.

(3) Regarding the radar analysis with P3 data: the authors have used a grid spacing of 1 km in the horizontal direction. With the fore/aft scanning technique and antenna rotation rate of 10 RPM in 2016 data, there is really no way to arrive at 1 km horizontal grid spacing for the wind analysis. The spacing of radials is on the order of ~ 1.4 km so ~ 2 km horizontal grid spacing is about as good as it gets. The authors would need to redo their analysis with ~ 2 km grid spacing and scale the ground-based analysis accordingly.  This could change some of the results. In addition, the storm core passed through on the far edge of the ground-based radar coverage when the P3 data was compared. The beam spread at this far range is

substantial and the grid spacing of the ground-based analysis of 1 km in the horizontal and 0.5 km in the vertical may not accurately reflect the pulse volume. Some discussion and/or analysis of this effect is also needed.

(4) Heavy use of NOAA TDR data is used in this study. This data is collected and processed (i.e., level 1 data) by NOAA and HRD. Unfortunately, the authors have not either (1)included a co-author from HRD or (2)made any mention of HRD in the acknowledgements section of the paper. NOAA/HRD has a data policy statement about these kinds of things that requires at the least, an acknowledgement for use of the TDR data. The authors need to rectify this in a re-submission of the paper.

Additional:
Lines 10 – 11; "A comparison between the two techniques shows that the axisymmetric tangential winds are generally comparable between the two techniques after the improvements to GVTD retrievals."  The comparisons of WV#0 tangential winds are generally comparable **before** the GVTD improvements as well. The improvements don't change the values that much. Sentence needs re-wording.

Line 21; don't need measurements from two or more radars if the platform is in motion, such as airborne Doppler radar. Please clarify.

Line 25; There are some papers that have analyzed wind retrieval techniques for TCs and with varying platforms. Please cite some of those papers here.

Line 27; I highly doubt that this is the first study to compare ground-based and airborne wind retrievals. The NOAA ground-based and airborne radars have been around for decades!

Lines 56 – 57; sentence doesn't read right, "…only a small portion of TC…"?  Please re-write.

General comment on the writing; at several places in the paper the word "the" needs to be inserted. Go through the paper again and look for these.
Some examples:
Line 96, "…and P3 TDR…" needs a 'the' before 'P3'
Line 98, "…of KAMX radar…" needs a 'the' before 'KAMX'

Line 122, Does this "mean" wind have the hurricane removed?

Equation (4), The two angles in the second terms on the RHS of (4) should be THETAt, not THETA.  This is probably just a typo.

Line 150, what is the lowest elevation angle used here and what error does this incur? The method must only work where cos(phi) ~ 1, so lowest scan level only.

Lines 177 – 179, this assumption is probably only valid above the boundary layer and below the outflow layer. Radial wind asymmetries can be substantial in the boundary layer. Some discussion of this is needed.

Page 8, this entire page could be significantly shortened because the "dynamic" centers are ultimately used, not the GBVTD centers. This can be summarized briefly.

Discussion regarding Figure 4 on page 9: I would say the results are mixed on the improvement of the "optimal solution" over the "original solution". For example, in one pass the green dashed line looks better than the blue dashed line, but in another pass, it looks worse and in the other passes the differences are negligible. Similar things for the orange and red lines.

Also, on page 9: what are the heights of comparison between the TDR and the 88D?  Since the storm core is on the far edge of the 88D coverage, the beam heights are probably fairly high, and the vertical velocities could be significant in this region. What is the impact of significant vertical velocity in the hurricane core on the TDR and 88D comparisons, given that the 88D retrievals don't take this into account?

Lines 265 – 268, These improvements are quite small, and I am wondering if they are statistically significant? I think the authors are overstating the impact of the improvements to the GVTD technique here and some rewording is needed.

Discussion around lines 295 – 296: these comparisons have significant differences between Ao and A1 coefficients in the eyewall region and it is not fair to say that they are "roughly consistent". Deviations of 2 m/s or less are a major error for Ao and A1 coefficients that have small values.

Lines 303 – 308, please see major comment (1).

Table 3, The differences between the original and improved GVTD method are only 0.35 m/s and this difference is likely not statistically significant. See major comment (2).

Table 4, I don't understand the wavenumber magnitudes listed here. Why is the magnitude of WV#0 so low? This should be azimuthal mean, correct? There is some confusion in the naming conventions listed in the table and the text that needs fixing.

Figure 6, should label these figures with the corresponding physical wind components because it is hard to follow.

---

## Referee Comment (RC2) · Anonymous Referee #2 · 30 Nov 2020

recommendation: I only found one thing to change, in the Introduction, noted below.Otherwise this a fine original paper which is perfectly suitable for publication in AMT. My comments below are mainly for the authors as I have no minor or major revisions to ask for.

Abstract: The abstract is a clear and concise description of the paper. Fine as is.

1. Introduction: You write " One limitation is that a ground-based Doppler radar has to be located outside of the radial distance between the radar and the storm center in order to sample the full tangential component of the vortex circulation accurately." I think you meant to write that the radar has to be outside the radius of maximum wind.

[Figure]

It is impossible for the radar to be located farther away than the storm center to radar distance!

2. Data Sets and Methodology: This section nicely summarizes the data procedures. I am curious why the SAMURAI analyses were interpolated to a polar array while the VoRTRAC were apparently still in Cartesian form. If in fact the end result of the Vortrac analyses also ended up in polar form that should be stated here.

3. The GVTD technique improvement: 3.1: a lot of math shown here but all of it necessary. I wonder if there was earlier work on data gaps and noise influence on maximum wavenumber to retrieve. Probably not needed but I believe the code in GBVTD to select max wavenumber was developed originally by the late Tom Matejka for the Extended VAD which is an ancestor of the VTD. Well, probably too much detail for this paper anyway. 3.2 GVTD-simplex center finding The description of the simplex minimization algorithm is a bit sparse. Maybe you should add a direct reference. Either that or remove the references to "contraction or expansion of the simplex" since you did not really explain what those terms mean. Also in the text when you mention comparisons of simplex-derived centers from different WSR88D radars you might list the locations: KMLB (Melbourne,FL) and KJAX (Jacksonville, FL). I think Harasti et al.(year?) had a nice description of many different of finding circulation centers. Also you might mention that the "dynamic"centers are smooth because the spline fit is to only a few aircraft derived centers, One per pass, so one would not expect them to show as much variation as the G*VTD centers available every 5-6 minutes. also you have an independent wind field from SAMURAI TDR analyses. Did you try simplex center finding on the pseudo-dual Doppler arrays? The Marks used the simplex to show variation of center with height in his airborne radar work and this was then later applied by Lee and Marks to the GBVTD.

4 Wind retrievals comparison between single Doppler and airborne dual Doppler analyses This is the real meat of this paper and it is amazing that no one did this before. Maybe it had to wait for the development of GEVTD and SAMURAI before such a study

could be done, but I think this part alone should be part of any class on radar meteorology from now on. 4.1 Wavenumber 0 tangential wind retrieval this section shows that GEVTD (both versions) capture well the wave 0 tangential wind by comparing with the pseudo dual Doppler analysis rom the P3. This is a slick idea to generate 88d obs by projecting the TDR analysis on the 88d radials and then doing the VTD on those "data" rather than on the original 88d data. As i read this, I couldn't help thinking you could just as easily plopped some idealized vortex on Matthews position and used totally synthetic data to test the VTD algorithm performance. I am not suggesting you do this as you already have the SAMURAI analysis, just a thought that you did not need a "real" storm to do this part of the analysis. 4.2 Asymmetric wind retrievals This is a very interesting discussion. I do wonder if the authors could say a bit more about the signature of the VRW in the reflectivity or is the VRW only visible in the windfield? It is easy to visualize in my mind the wave 1 asymmetry, but higher wave numbers make me think there should be a series of bumps in the dBZ, or is the reality that the asymmetries are more finer in scale, so requiring higher wave numbers? That is, are convective cells in the RMW giving rise to wind asymmetries that are then aliased onto wave #2? well, probably can't answer that with this dataset. I know that VORTRAC also can produce a MSLP estimate from the symmetric wind field. I wonder if the VTD analyses give mslp similar to the flight level data. If the wind fields agree so well I would think the pressure retrievals should also. Not necessary for this paper though. 5 Conclusions the authors nicely summarize their work here and hint that there is more to be said about the analyses themselves. In this paper they have validated both methods of analysis and I imagine the next Cha et al. will go into more detail about the VRW's and other features of this dataset including the eyewall replacement. This methods-oriented paper is a fine introduction to that topic.

---

## Referee Comment (RC3) · Anonymous Referee #3 · 10 Dec 2020

Review of "Comparison of Single Doppler and Multiple DopplerWind Retrievals in Hurricane Matthew (2016)"

**General comments**:

This paper evaluates the accuracy of the generalized velocity track display (GVTD) technique by comparing the wind field obtained from the airborne tail Doppler radar (TDR) data. The evaluation of the GVTD technique in a real case has not been done so far and has been desired. Additionally, this paper re-derives the GVTD technique to obtain a more accurate wind field. Generally speaking, it is hard to compare the difference between observations from different measurements because it is necessary to consider the strengths and weaknesses of each observing capability. The authors did a great job working on this difficulty by carefully looking at the retrieved wind field.

This paper is well written, and the purpose and results of this study are clear. Although I have some questions to better understand the GVTD technique, I recommend acceptance once the authors address the questions.

**Recommendation:** Minor revisions

**Specific comments:**

L180: Here, I'd like to make sure which variables can actually be retrieved from Eqs. 15-19. The sentence describes that the GVTD provides the along-beam component of the mean flow (i.e., Eq. 15), axisymmetric tangential wind (i.e., Eq. 16), axisymmetric radial wind (i.e., Eq. 17), and asymmetric tangential winds (n=1-2) (i.e., Eqs. 18 and 19). But, how can we obtain axisymmetric radial wind and the along-beam component of the mean flow? Eq. 15 includes axisymmetric radial wind on the right hand side and Eq. 17 includes the along-beam component of the mean flow on the right hand side. Rearranging Eqs. 15 and 17 is needed to obtain axisymmetric radial wind and the along-beam component of the mean flow.

Here, $V_{M\|}$ indicates $V_M \cos ()$, $\alpha$ indicates $R/R_T$, and the storm motion is assumed to be zero.

Eq. 15: $V_{M\|} = A0 - \alpha*VR\_C0 + 1/2*VT\_S1$

Eq: 17: $VR\_C0 = (A0+A1+A2+A3+A4)/(1+\alpha) - V_{M\|}$

$VT\_S1 = 2*A2 + 2*A4$ (from Eq. 18)

Substituting Eq. 17 into Eq. 15 yields

$V_{M\|} = A0 - \alpha*((A0+A1+A2+A3+A4)/(1+\alpha) - V_{M\|}) + A2 + A4$

$(1-\alpha) * V_{M\|} = A0 + A2 + A4 - \alpha*(A0+A1+A2+A3+A4)/(1+\alpha)$

Then, $V_{M\parallel} = (A0+A2+A4)/(1-\alpha) - \alpha*(A0+A1+A2+A3+A4)/(1-\alpha^2)$

Substituting $V_{M\parallel}$ into Eq. 17 yields

$VR\_C0 = (A0+A1+A2+A3+A4) *(1-\alpha)/(1-\alpha^2) - (A0+A2+A4)/(1-\alpha)$
$\qquad - \alpha*(A0+A1+A2+A3+A4)/(1-\alpha^2)$

Then, $VR\_C0 = - (A0+A2+A4)/(1-\alpha) + (A0+A1+A2+A3+A4)/(1-\alpha^2)$

Is this derivation wrong?

Additionally, I wonder why the authors don't evaluate the accuracy of axisymmetric radial wind in this study. If we find that both axisymmetric tangential and radial winds can be retrieved from the GVTD technique with acceptable accuracy, then the GVTD-retrieved winds can be useful for diagnosing a possibility of changes in storm size.

L218: This is just a comment. In my experience, a method from Bell and Lee (2012) provides better centers in terms of time consistency. That being said, I understand that you use the dynamic centers.

L225: Specify the mean wind component is an unknown variable or a given value? If it is a given value, how did the authors obtain it?

Figs. 3a and b: KAMX is not located at the center of the figure.

Figs. 3c and d: There is no caption about these figures.

L303: This description appears to me that the problem comes from the airborne "dual" Doppler analysis method, which essentially uses the fore/aft scanning technique. However, I don't think that the problem here is from the steady-state assumption in the dual Doppler analysis. As described in section 1, the forward and aft scannings are conducted within a few seconds, allowing for a nearly simultaneous observation, at least, near the aircraft (thus, the steady-state assumption is valid). I thought the problem here is that retrieved wind vectors observed at different times within ~45 min are synthesized into one picture in SAMURAI software, assuming that they are steady-state (e.g., Fig. 3b). Is my understanding wrong?

---

## Author Comment (AC1) · 8 Jan 2021

Authors are grateful to the reviewer for a careful reading of the work and constructive comments. The responses are in the attached file

Please also note the supplement to this comment: https://amt.copernicus.org/preprints/amt-2020-240/amt-2020-240-AC1-supplement.pdf

---

## Author Comment (AC2) · 8 Jan 2021

**amt-2020-240**

Thank you for reviewing the manuscript and providing constructive comments. We have made edits to the manuscript incorporated with your suggestions. Reviewers' comments are shown in black, our response to each comment is shown in blue, and changes to the manuscript are shown in red.

**Reviewer #2:**

Recommendation: I only found one thing to change, in the Introduction, noted below. Otherwise this a fine original paper which is perfectly suitable for publication in AMT. My comments below are mainly for the authors as I have no minor or major revisions to ask for.

Abstract: The abstract is a clear and concise description of the paper. Fine as is.

Thank you for the comment.

1. Introduction: You write " One limitation is that a ground-based Doppler radar has to be located outside of the radial distance between the radar and the storm center in order to sample the full tangential component of the vortex circulation accurately." I think you meant to write that the radar has to be outside the radius of maximum wind. It is impossible for the radar to be located farther away than the storm center to radar distance!

Thank you for the comment. We have revised the sentence.

One limitation is that the radial distance between the radar and the storm center has to be large enough to sample the tangential component of the vortex circulation in order to minimize the geometric distortion.

2. Data Sets and Methodology: This section nicely summarizes the data procedures. I am curious why the SAMURAI analyses were interpolated to a polar array while the VoRTRAC were apparently still in Cartesian form. If in fact the end result of the Vortrac analyses also ended up in polar form that should be stated here.

Thank you for the comment. We have added a comment to the manuscript to clarify.

The gridded data was further analyzed using the Vortex Objective Radar Tracking and Circulation (VORTRAC) software in LROSE to interpolate onto a cylindrical coordinate and obtain the kinematic structure by the improved GVTD algorithm formulated in section 3.1.

3. The GVTD technique improvement: 3.1: a lot of math shown here but all of it necessary. I wonder if there was earlier work on data gaps and noise influence on maximum wavenumber to retrieve. Probably not needed but I believe the code in GBVTD to select max wavenumber was developed originally by the late Tom Matejka for the Extended VAD which is an ancestor of the VTD. Well, probably too much detail for this paper anyway.

Thank you for the comment. We have added the reference of Matejka and Srivastava 1991.

To deal with missing data in observational radar data and reduce the influence of outliers [Matejka and Srivastava, 1991], the truncation of the Fourier series follows Lee et al. (2000) (Table 2), which is consistent with the restriction of maximum allowable gap size in Lorsolo and Aksoy (2012).

3.2 GVTD-simplex center finding The description of the simplex minimization algorithm is a bit sparse. Maybe you should add a direct reference. Either that or remove the references to "contraction or expansion of the simplex" since you did not really explain what those terms mean. Also in the text when you mention comparisons of simplex-derived centers from different WSR88D radars you might list the locations: KMLB (Melbourne,FL) and KJAX (Jacksonville, FL). I think Harasti et al.(year?) had a nice description of many different of finding circulation centers.

Thank you for the comment. We have added the reference of Harasti et al. 2004.

The simplex center is found by maximizing the mean tangential wind within an axisymmetric TC with three operations on a simplex: reflection, contraction, and expansion [Lee and Marks, 2000, Harasti et al., 2004].

Also you might mention that the "dynamic" centers are smooth because the spline fit is to only a few aircraft derived centers, One per pass, so one would not expect them to show as much variation as the G\*VTD centers available every 5-6 minutes. also you have an independent wind field from SAMURAI TDR analyses.

Thank you for the comment. We have clarified the sentence.

The centers are interpolated from a few dynamic centers with a series of spline curves every two minutes, so the centers are connected into a continuous track.

Did you try simplex center finding on the pseudo-dual Doppler arrays? The Marks used the simplex to show variation of center with height in his airborne radar work and this was then later applied by Lee and Marks to the GBVTD.

Thank you for the comment. We have tried the simplex center finding on the pseudo-dual Doppler analysis, and the discrepancies between the simplex centers and dynamic centers are small. The retrieved wind fields are similar, as well as the asymmetric structure. Therefore, we decide to use the dynamic centers to be consistent with our future work discussing Hurricane Matthew's asymmetric structure observed by the ground-based radars.

4 Wind retrievals comparison between single Doppler and airborne dual Doppler analyses: This is the real meat of this paper and it is amazing that no one did this before. Maybe it had to wait for the development of GEVTD and SAMURAI before such a study could be done, but I think this part alone should be part of any class on radar meteorology from now on.

Thank you very much for the positive endorsement!

4.1 Wavenumber 0 tangential wind retrieval this section shows that GEVTD (both versions) capture well the wave 0 tangential wind by comparing with the pseudo dual Doppler analysis from the P3. This is a slick idea to generate 88d obs by projecting the TDR analysis on the 88d radials and then doing the VTD on those "data" rather than on the original 88d data. As I read this, I couldn't help thinking you could just as easily plopped some idealized vortex on Matthews position and used totally synthetic data to test the VTD algorithm performance. I am not suggesting you do this as you already have the SAMURAI analysis, just a thought that you did not need a "real" storm to do this part of the analysis.

Thank you for the comment. In response to this and another reviewer's comment, we have done an idealized experiment to show that the steady-state assumption of one straight flight leg to synthesize the data into one snapshot with the presence of rapidly evolving features could result in temporal aliasing and unrealistic asymmetric component retrievals. We have added a new subsection 4.3 and Figure 8 (Fig. 1 here) in the revised manuscript.

4.2 Asymmetric wind retrievals This is a very interesting discussion. I do wonder if the authors could say a bit more about the signature of the VRW in the reflectivity or is the VRW only visible in the wind field? It is easy to visualize in my mind the wave 1 asymmetry, but higher wave numbers make me think there should be a series of bumps in the dBZ, or is the reality that the asymmetries are more finer in scale, so requiring higher wave numbers? That is, are convective cells in the RMW giving rise to wind asymmetries that are then aliased onto wave 2? well, probably can't answer that with this dataset.

Thank you for the comment. Cha et al. 2020 shows the signature of the VRW in the reflectivity and the GVTD-retrieved tangential wind fields. The propagation speeds of asymmetric tangential wind and reflectivity signals are consistent with the linear wave theory, suggesting that the observed asymmetries are well described by VRW theory (Fig. 4 in Cha et al. 2020).

Figure 2 shows the azimuthal temporal evolution of wavenumber 1 and 2 of reflectivity and tangential wind in the inner eyewall region of Hurricane Matthew. The wavenumber 1 reflectivity is stationary from 1915 to 1945 UTC, whereas the wavenumber 2 reflectivity signals are also propagating, similar to the wavenumber 2 tangential wind signals. We have added the features description of the wavenumber 1 and 2 reflectivity signals in the manuscript, but the plot of reflectivity is not shown.

To test the hypothesis, the phases of maximum wavenumber 1 and wavenumber 2 tangential winds retrieved from the 5-minute single Doppler observations are examined in Fig. 7 for the temporal evolution during the first flight pass from 1907 to 1940 UTC. The amplitude and phase (Eqs. 18 and 19) of wavenumber 1 and 2 tangential winds are denoted in polar coordinates by the radius and azimuth, respectively. The wavenumber 1 tangential wind (Fig. 7a) generally stayed unchanged throughout the first pass with a magnitude between 8 and 12 m s-1 and phase to the E to NE (same as the wavenumber 1 reflectivity, not shown here). Environmental vertical wind shear derived from the Statistical Hurricane Intensity Prediction Scheme dataset (SHIPS) points to the northeast direction with a magnitude of 7 m s-1, suggesting that the wavenumber 1 distribution is forced by the vertical wind shear to be consistently in the downshear-right quadrant.

The wavenumber 2 tangential wind (Fig. 7b) propagated cyclonically during the flight pass (same as the wavenumber 2 reflectivity, not shown here) with a magnitude up to 7 m s-1. The propagation of the wavenumber 2 tangential wind is estimated to be 285

degrees from 1907 to 1940 UTC, which is 35 m s-1, or 63% of  $V_{Tmax}$ . The propagation of the wavenumber 2 tangential wind is roughly consistent with linear VRW theory [Kuo et al., 1999, Cha et al., 2020].

I know that VORTRAC also can produce a MSLP estimate from the symmetric wind field. I wonder if the VTD analyses give mslp similar to the flight level data. If the wind fields agree so well I would think the pressure retrievals should also. Not necessary for this paper though.

Thank you for the comment. We have computed the perturbation pressure deficit using the gradient wind balance equation. The integrated perturbation pressure deficit retrievals from different methods agree well with the 'truth'. The analysis and RMS difference of the integrated perturbation pressure deficit are added to Table 4 in the manuscript.

The perturbation pressure deficit is integrated from r = 10 to 70 km using the gradient wind balance equation [Lee et al., 2000]. The integrated perturbation pressure deficit retrievals from different methods agree well with the "truth" (~ 1 mb RMS in general). Similar as the results of RMS difference of  $V_TC_0$ , including the storm motion terms decreases the RMS differences about 0.2 mb. The RMS difference of improved GVTD algorithm from the single Doppler retrieval has the least deviation from the "truth", suggesting that the perturbation pressure deficit derived from the single Doppler observations has high fidelity.

5 Conclusions the authors nicely summarize their work here and hint that there is more to be said about the analyses themselves. In this paper they have validated both methods of analysis and I imagine the next Cha et al. will go into more detail about the VRW's and other features of this dataset including the eyewall replacement. This methods-oriented paper is a fine introduction to that topic.

Thank you for the comment. Some additional analyses using the improved formulation can be found in Cha (2018), with continued analysis submitted to the peer-reviewed literature in the future.

---

## Author Comment (AC3) · 8 Jan 2021

amt-2020-240

Thank you for reviewing the manuscript and providing constructive comments again. We have made edits to the manuscript incorporated with your suggestions. Reviewers' comments are shown in black, our response to each comment is shown in blue, and changes to the manuscript are shown in red.

Reviewer #3:
Review of "Comparison of Single Doppler and Multiple DopplerWind Retrievals in Hurricane Matthew (2016)" General comments: This paper evaluates the accuracy of the generalized velocity track display (GVTD) technique by comparing the wind field obtained from the airborne tail Doppler radar (TDR) data. The evaluation of the GVTD technique in a real case has not been done so far and has been desired. Additionally, this paper re-derives the GVTD technique to obtain a more accurate wind field. Generally speaking, it is hard to compare the difference between observations from different measurements because it is necessary to consider the strengths and weaknesses of each observing capability. The authors did a great job working on this difficulty by carefully looking at the retrieved wind field. This paper is well written, and the purpose and results of this study are clear. Although I have some questions to better understand the GVTD technique, I recommend acceptance once the authors address the questions.
Recommendation: Minor revisions

Specific comments:

L180: Here, I'd like to make sure which variables can actually be retrieved from Eqs. 15-19. The sentence describes that the GVTD provides the along-beam component of the mean flow (i.e., Eq. 15), axisymmetric tangential wind (i.e., Eq. 16), axisymmetric radial wind (i.e., Eq. 17), and asymmetric tangential winds (n=1-2) (i.e., Eqs. 18 and 19). But, how can we obtain axisymmetric radial wind and the along-beam component of the mean flow? Eq. 15 includes axisymmetric radial wind on the right hand side and Eq. 17 includes the along-beam component of the mean flow on the right hand side. Rearranging Eqs. 15 and 17 is needed to obtain axisymmetric radial wind and the along-beam component of the mean flow. Here, VM|| indicates VM cos (), $\alpha$ indicates R/RT, and the storm motion is assumed to be zero. Eq. 15:

$$V_M \parallel = A_0 - \frac{R}{R_T}V_R C_0 + \frac{1}{2}V_T S_1 - \frac{1}{2}V_R C_1 - U_S cos\theta_T - V_S sin\theta_T$$

$$V_R C_0 = \frac{A_0 + A_1 + A_2 + A_3 + A_4}{1 + \frac{R}{R_T}} - V_M cos(\theta_T - \theta_M) - V_R C_1 - V_R C_2 - V_R C_3 - \frac{R}{R_T}(U_S cos\theta_T + V_S sin\theta_T)$$

Substituting Eq. 17 into Eq. 15 yields VM|| = A0 − $\alpha$ × ((A0+A1+A2+A3+A4)/(1+$\alpha$) − VM|| ) + A2 + A4 (1−$\alpha$) × VM|| = A0 + A2 + A4 − $\alpha$ × (A0+A1+A2+A3+A4)/(1+$\alpha$) Then, VM|| = (A0+A2+A4)/(1−$\alpha$) − $\alpha$ × (A0+A1+A2+A3+A4)/(1−$\alpha^2$) Substituting VM|| into Eq. 17 yields VRC0 = (A0+A1+A2+A3+A4) *(1−$\alpha$)/(1−$\alpha^2$ ) − (A0+A2+A4)/(1−$\alpha$) − $\alpha$ × (A0+A1+A2+A3+A4)/(1−$\alpha^2$ )
Then, VRC0 = − (A0+A2+A4)/(1−$\alpha$) + (A0+A1+A2+A3+A4)/(1−$\alpha^2$ ) Is this derivation wrong?

Thank you for the comment. We have derived the $V_R C_0$ term to the same formula independently, but did not include that in the original manuscript. The $V_M cos(\theta_T - \theta_M)$ term can be further derived after the computation of $V_R C_0$. We have added the equation to the manuscript.

Plugging Eq. 16 into Eq. 15 to derive Eq. 20:

$$V_R C_0 = \frac{A_0 + A_1 + A_2 + A_3 + A_4}{(1 - \frac{R^2}{R_T{}^2})} - \frac{A_0 + A_2 + A_4}{(1 - \frac{R}{R_T})}$$

$$- V_R C_2 - \frac{\frac{R}{R_T}}{1 - \frac{R}{R_T}} V_R C_4 - \frac{1}{2}(\frac{1}{1 - \frac{R}{R_T}})(V_T S_5 - V_R C_5)$$

Equations 15 - 19 correspond to equations (16)-(20) in [Jou et al., 2008] with the additional terms of storm motion on Eqs. 15 - 17. Equation 20 is an updated version of Eq. 15 to minimize the unknown terms after plugging in the $V_M cos(\theta_T - \theta_M)$.

Additionally, I wonder why the authors don't evaluate the accuracy of axisymmetric radial wind in this study. If we find that both axisymmetric tangential and radial winds can be retrieved from the GVTD technique with acceptable accuracy, then the GVTD-retrieved winds can be useful for diagnosing a possibility of changes in storm size.

Thank you for the comment. As shown in the manuscript, the retrievals of $A_2$, $A_3$ are variable due to the propagation of wavenumber 2 winds. Since the axisymmetric radial wind is influenced by the harmonic 2 and 3 components, we cannot fully validate the axisymmetric radial wind retrieval. We have added a comment to the manuscript.

Since the axisymmetric radial wind is influenced by the harmonic 2 and 3 components (Eq. 20), we cannot fully validate the axisymmetric radial wind retrieval with the current dataset. The evaluation for the accuracy of the axisymmetric radial wind retrieval is not included in this study.

L218: This is just a comment. In my experience, a method from Bell and Lee (2012) provides better centers in terms of time consistency. That being said, I understand that you use the dynamic centers.

Thank you for the comment. We have tested the objective centers using Bell and Lee 2012 method, but the results are not optimal. The simplex centers are variable, so the objective method cannot get the best fit of the centers in this case.

L225: Specify the mean wind component is an unknown variable or a given value? If it is a given value, how did the authors obtain it?

Thank you for the comment. The mean wind magnitude and direction were derived from the airborne dual Doppler analysis, following the procedure proposed by [Marks et al., 1992]. The storm-relative horizontal wind field ($V_r$)in a cylindrical coordinate system centered on the storm can be decomposed into:

$$V_r(r, \theta, z) = \bar{V}_r(z) + V'(r, \theta, z) \tag{1}$$

where r is radius, $\theta$ is azimuth, z is height, $\bar{V}_r(z)$ is the horizontally averaged wind vector over the radius and azimuth, and $V'(r, \theta, z)$ is the deviation from $\bar{V}_r(z)$. $\bar{V}_r(z)$ can be expressed as:

$$\bar{V}_r(z) = \frac{1}{2\pi} \int_0^{2\pi} \int_0^{r_{max}} V_r(r, \theta, z) dr d\theta \tag{2}$$

If the horizontal wind field is from a circular symmetric vortex with no steering flow, $\bar{V}_r(z)$ would be zero. Nevertheless, if the vortex is embedded in the steering flow, the averaged horizontal wind field would equal the mean wind component. Thus, the local wind shear can be approximated by subtraction of the mean wind component at different altitudes. In our study, we calculated the mean wind component averaged from the vortex inner core area within the radius of 60 km. We have clarified how we derived the mean wind component in the manuscript.

The mean wind $(V_M)$ is the horizontal average of the environmental flow at each altitude following the procedure proposed by [Marks et al., 1992], which can be used to calculate the vertical wind shear.

Figs. 3a and b: KAMX is not located at the center of the figure.
Figs. 3c and d: There is no caption about these figures.

Thank you for the comment. We have revised the caption.

Doppler velocity at z = 4 km (a) observed by KAMX radar at 1921 UTC, and (b) resampled from dual Doppler analysis synthesized from 1855 - 1940 UTC. Reflectivity at z = 4 km (c) observed by KAMX radar and (d) derived from dual Doppler analysis. The timing of (c) and (d) are the same as (a) and (b), respectively. The black star denotes the TC center, and the dashed circle denotes the radius of maximum wind of 18 km.

L303: This description appears to me that the problem comes from the airborne "dual" Doppler analysis method, which essentially uses the fore/aft scanning technique. However, I don't think that the problem here is from the steady-state assumption in the dual Doppler analysis. As described in section 1, the forward and aft scannings are conducted within a few seconds, allowing for a nearly simultaneous observation, at least, near the aircraft (thus, the steady-state assumption is valid). I thought the problem here is that retrieved wind vectors observed at different times within 45 min are synthesized into one picture in SAMURAI software, assuming that they are steady-state (e.g., Fig. 3b). Is my understanding wrong?

Thank you for the comment. This is an excellent point. We have now clarified that there are in effect two different types of sampling errors that arise from the steady state assumption – the first is due to the time lag between fore and aft beams, and the second is due to the length of time used to composite the multi-Doppler into a single snapshot. While both produce some errors, the latter is more consequential when considering low-wavenumber asymmetries since it takes longer to capture the larger scale structure, resulting in evolution over the flight pass. The 'local' wind may be correct, but the overall structure is distorted by collapsing to a single time. We hypothesize that the discrepancies of retrieved wavenumber 1 and 2 tangential winds are due to the steady state assumption in the dual Doppler wind synthesis into one snapshot, not the fore/aft lag.

We hypothesize that the discrepancies of retrieved wavenumber 1 and 2 tangential winds are due to the steady state assumption in the dual Doppler wind synthesis into one snapshot. Two different types of sampling errors in effect arise from the steady state assumption – the first is due to the time lag between fore and aft beams, and the second is due to the length of time used to composite the multi-Doppler into a single snapshot. While both produce some errors, the latter is more consequential when considering the temporal evolution of the phenomena is faster than the period of data collection, resulting in evolution over the flight pass. The "local" wind may be correct, but the overall structure is distorted by collapsing to a single time. For example, the propagation velocity of a wavenumber 2 vortex Rossby wave (VRW) is half of the symmetric tangential wind velocity [Lamb, 1932, Guinn and Schubert, 1993, Kuo et al., 1999]. A propagating wavenumber 2 asymmetry could then alias onto other wavenumbers, contributing to a discrepancy in wavenumber 1 tangential wind.

**References**

[Bell and Lee, 2012] Bell, M. M. and Lee, W.-C. (2012). Objective tropical cyclone center tracking using single-Doppler radar. *Journal of Applied Meteorology and Climatology*, 51(5):878–896.

[Guinn and Schubert, 1993] Guinn, T. A. and Schubert, W. H. (1993). Hurricane spiral bands. *Journal of the Atmospheric Sciences*, 50(20):3380–3403.

[Jou et al., 2008] Jou, B. J.-D., Lee, W.-C., Liu, S.-P., and Kao, Y.-C. (2008). Generalized VTD retrieval of atmospheric vortex kinematic structure. Part I: Formulation and error analysis. *Mon. Wea. Rev.*, 136(3):995–1012.

[Kuo et al., 1999] Kuo, H.-C., Williams, R., and Chen, J.-H. (1999). A possible mechanism for the eye rotation of Typhoon Herb. *J. Atmos. Sci.*, 56(11):1659–1673.

[Lamb, 1932] Lamb, H. (1932). *Hydrodynamics.* Cambridge university press.

[Marks et al., 1992] Marks, F. D., Houze, R. A., and Gamache, J. F. (1992). Dual-aircraft investigation of the inner core of Hurricane Norbert. Part I: Kinematic structure. *J. Atmos. Sci.*, 49(11):919–942.

---

## Author Comment (AC4) · 8 Jan 2021

amt-2020-240

Thank you for reviewing the manuscript and providing constructive comments. We have made edits to the manuscript incorporated with your suggestions. Reviewers' comments are shown in black, our response to each comment is shown in blue, and changes to the manuscript are shown in red.

Reviewer #1:
There are some interesting and useful components of this manuscript related to differences in ground-based and airborne wind retrieval methods. The writing and organization are clear, for the most part. However, there are several major concerns with the paper as listed below that will take a significant amount of time to address. I therefore recommend that the paper be declined and re-submitted once these issues are addressed.

Major:
(1) The total error in the radar comparisons is a summation of instrument effects (e.g., signal-to-noise ratio), sampling effects (e.g., gaps in data coverage, spatial/temporal resolution) and algorithm effects (e.g., geometry, approximations, solution method). The comparisons between dual Doppler and single Doppler retrievals for WV#0 tangential winds have some differences in the eyewall region (especially pass 4) that appear to be due to differences in the GVTD method (algorithm effect). The authors are trying to isolate the effects of the "steady-state" assumption (sampling effect) in airborne radar analysis by comparing WV#1 and WV#2 tangential winds from dual Doppler and single Doppler retrievals. However, it is not at all clear how much of the differences the authors are seeing are due to instrument effects, algorithm effects or other sampling effects. The WV#1 and WV#2 tangential winds should have larger errors due to algorithm effects when compared to the WV#0 tangential winds. This makes it very difficult or impossible to isolate the effects of just the steady-state assumption and thus the conclusions from this analysis are uncertain. The authors need to isolate these effects through some type of simulated analysis in order to make definitive conclusions.

Thank you for the comment. We agree we did not adequately discuss the total error in the radar comparisons in the manuscript. Additional discussion, previous literature reviews of the sources of error, and a new figure supporting our claim that the sampling error is a large contributor to the differences have been added to the manuscript. The instrument and algorithm errors are relatively small, but the sampling effect, particularly a long period of data collection for the airborne Doppler radar observation, could result in aliasing in the analysis with the steady-state assumption in a snapshot. To provide further support for this hypothesis, we performed an idealized experiment with a rotating wavenumber 2 tangential wind sampled by an aircraft with a realistic scanning strategy. We 'fly' a simulated aircraft track through an idealized vortex in a typical straight-line flight track during P3 operational reconnaissance. A wavenumber 2 Rankine edge wave propagating around 20 km RMW at $1/2\ V_{max}$ is sampled by the simulated flight, which takes 22 minutes to finish a 160 km flight leg.

The new Figure 8 in the manuscript, reproduced here (Fig. 1), shows the derived tangential wind for a rotating wavenumber 2 during the flight pass. In this idealized experiment, we have minimized all the other errors such that a steady-state vortex is retrieved nearly

[Figure]

Figure 1: (a) Idealized dual-Doppler tangential wind speed retrieved from a straight flight pass through propagating wavenumber two asymmetry (color in m s$^{-1}$) and 50 m s$^{-1}$ contour of time-averaged wavenumber two in black. The initial phase of the propagating wavenumber 2 tangential wind is oriented from east to west and final phase is from north to south. (b) Wavenumber 1 plus aircraft flight track from south to north, (c) Wavenumber 2, and (d) Wavenumber 3 tangential wind components retrieved by the Fourier decomposition of the tangential wind field shown in panel (a).

exactly. The distorted tangential wind field suggests temporal aliasing from the extended sampling period of the aircraft pass due to the propagation of the wavenumber 2 asymmetry. Although we only prescribed a propagating wavenumber 2 Rankine wave in the experiment, both wavenumber 1 and 3 components are with non-trivial amplitude. Based on comments from another reviewer, we have also clarified that there are in effect two different types of sampling errors that arise from the steady state assumption – the first is due to the time lag between fore and aft beams, and the second is due to the length of time used to composite the multi-Doppler into a single snapshot. While both produce some errors, the latter is more consequential when considering low-wavenumber asymmetries since it takes longer to capture the larger scale structure, resulting in evolution over the flight pass. This idealized experiment shows that the steady-state assumption of one straight flight leg to

synthesize the data into one snapshot with the presence of rapidly evolving features could result in temporal aliasing and result in potential misinterpretation of the asymmetric flow field. Additional errors are possible from instrument and algorithm effects, and this has now been clarified in the revised manuscript.

We have included the discussion on different sources of error here for reference. Please see the revised manuscript for the full description of the new Figure 8.

Although dual-Doppler observations can be used to assess snapshots of high resolution kinematic and convective structure, airborne reconnaissance and research missions are rare events in most of the countries impacted by TCs. The three-dimensional airflow structure can also be retrieved from the dual-Doppler observations when the system is detected by two ground-based radars. General sources of error in the inter-comparison of ground-based and airborne dual-Doppler observations include instrument effects, algorithm effects, and sampling effects [Hildebrand and Mueller, 1985]. Instrument effects include the effects of attenuation and signal-to-noise ratio that could be caused by the radar processor design or measurement technique. These effects are likely to be most influential with marginal signal-to-noise, but random velocity errors up to 1 m s$^{-1}$ are possible with many radar designs, including airborne radars [Hildebrand et al., 1994]. Algorithm effects include the effects of the interpolation to the Cartesian grid, multi-Doppler geometry, the solution method and its associated assumptions, and the derivation of the vertical velocities. Sampling effects include the effects of data spacing and density, geometry of flight tracks, temporal changes in the storm, advection, and data collection period. One of the long-lasting problems is the length of time required for each flight leg with airborne Doppler radar [Ray and Stephenson, 1990].The temporal effects can degrade the analysis if the data collection takes too long. [Jorgensen et al., 1983] quantitatively compared the wind fields of homogeneous precipitation derived from the two pseudo-orthogonal flight legs and two ground-based dual-Doppler observations. Their measurements showed agreement in the horizontal wind fields, but small discrepancies in the vertical velocities about 0.5 - 1 m s$^{-1}$ for the airborne system and about 0.2 m s$^{-1}$ for the ground-based system. The discrepancy was attributed to uncertainties in the pointing angle of the airborne system and a long data collection period.

(2) Regarding the improved GVTD method: the authors have done a thorough analysis of the impacts of storm motion on the ground-based retrieval. I think this analysis is useful for the community. However, I think the revised method only makes minor improvements on the errors in the retrieved mean tangential winds, but the authors have overstated their importance at several places in the paper, including the abstract. Thus, the tone of the paper needs to be revised in several places and more details are found in the additional comments section.

Thank you for the comment, and we have revised the tone throughout the paper. As the reviewer has mentioned above, the error sources can come from instrument effects, sampling effects, and algorithm effects. While the improvement is small, as discussed in Jorgensen et al. 1983 , the accumulation of different sources of errors can contribute to a large amount, such that a minimization of all the known errors is needed. The errors from the contribution of storm motion could be large if the storm moves fast or when the storm tracks closer to the radar ($R_T$ is small). The estimated storm motion of Hurricane Matthew (2016) is 8.4, 8.2, 3.8, 6.1 m s$^{-1}$ on pass 1 to 4, respectively, and Matthew is about 200 km away from the radar. Therefore, the averaged RMSE of the four passes shown in the manuscript is

not that large, but we think that the improvement on the errors from the improved GVTD technique are important and contribute to the reduction in total error. The accuracy of the estimated hurricane intensity is important when the storm moves inland or closer to the radar observation range. We have modified the tone of the description for the results, with two examples given here:

**Abstract:** A comparison between the two techniques shows that the axisymmetric tangential winds are generally comparable between the two techniques, and the improved GVTD technique improves the accuracy of the retrieval.

While it is a relatively small reduction in the RMS difference in the current case, the statistically significant difference in this algorithm error contributes to an overall reduction in the total error from instrument, algorithm, and sampling contributions.

**Conclusion:** A comparison between the two techniques shows that the retrieved axisymmetric (wavenumber 0) component of tangential winds are generally comparable between the two techniques, and the improved GVTD technique improves the accuracy of the retrieval.

(3) Regarding the radar analysis with P3 data: the authors have used a grid spacing of 1 km in the horizontal direction. With the fore/aft scanning technique and antenna rotation rate of 10 RPM in 2016 data, there is really no way to arrive at 1 km horizontal grid spacing for the wind analysis. The spacing of radials is on the order of 1.4 km so 2 km horizontal grid spacing is about as good as it gets. The authors would need to redo their analysis with 2 km grid spacing and scale the ground-based analysis accordingly. This could change some of the results. In addition, the storm core passed through on the far edge of the ground-based radar coverage when the P3 data was compared. The beam spread at this far range is substantial and the grid spacing of the ground-based analysis of 1 km in the horizontal and 0.5 km in the vertical may not accurately reflect the pulse volume. Some discussion and/or analysis of this effect is also needed.

Thank you for the comment. We realize we neglected to describe the fact that low-pass filtering was applied to the P-3 analysis to ensure that the resolved scales were in fact consistent with the data spacing, but this has now been corrected in the revised manuscript. As shown by Koch et al. 1983 and others, we believe the grid spacing should be smaller than the data spacing in order to accurately resolve the maximum spatial scales available for the given sampling. In SAMURAI, the 'grid spacing' is actually a 'nodal-spacing' since the resolved wind field is a function composed of finite elements. The nodal-spacing determines the minimum spatial scale resolved by the spline function of 2dx. In the current analyses, we use a 4dx Gaussian filter in the horizontal and 2dx filter in the vertical. Therefore, the resolved spatial scales are ∼4 km in the horizontal, which is well above both the along-track spacing, radar range volume, and azimuthal beam volume. While the vertical resolution may be a bit fine even after filtering, we focus on the horizontal structure in the paper and feel our analysis accurately resolves the appropriate spatial scales for the given sampling.

A low-pass filtering was applied to the P-3 analysis to ensure that the resolved scales were in fact consistent with the data spacing. The grid spacing should be smaller than the data spacing in order to accurately resolve the maximum spatial scales available for the given sampling [Koch et al., 1983].

(4) Heavy use of NOAA TDR data is used in this study. This data is collected and processed (i.e., level 1 data) by NOAA and HRD. Unfortunately, the authors have not either (1)included a co-author from HRD or (2) made any mention of HRD in the acknowledgements section of the paper. NOAA/HRD has a data policy statement about these kinds of things that requires at the least, an acknowledgement for use of the TDR data. The authors need to rectify this in a re-submission of the paper.

Thank you for the comment and we apologize for the oversight. We have added the acknowledgement of NOAA/HRD, and are very grateful for the data collection efforts of NOAA Aircraft Operations Center and HRD.

We would like to thank NOAA Aircraft Operations Center and the Hurricane Research Division of the Atlantic Oceanographic and Meteorological Laboratory for collecting the airborne tail Doppler radar data used for this study, and the National Weather Service for the ground-based radar data.

Additional:

Lines 10 – 11; "A comparison between the two techniques shows that the axisymmetric tangential winds are generally comparable between the two techniques after the improvements to GVTD retrievals." The comparisons of WV0 tangential winds are generally comparable before the GVTD improvements as well. The improvements don't change the values that much. Sentence needs re-wording.

Thank you for the comment. We have reworded the sentence.

A comparison between the two techniques shows that the axisymmetric tangential winds are generally comparable between the two techniques, and the improved GVTD technique improves the accuracy of the retrieval.

Line 21; don't need measurements from two or more radars if the platform is in motion, such as airborne Doppler radar. Please clarify.

Thank you for the comment. We have clarified the sentence.

In addition to the presence of an airborne Doppler radar with fore/aft capability or multiple radars with sufficient range and geometry around the TC, a steady state assumption during the Doppler radar observation period is required to synthesize the wind fields into one snapshot in time.

Line 25; There are some papers that have analyzed wind retrieval techniques for TCs and with varying platforms. Please cite some of those papers here.

Thank you for the comment. Several references have been added.

Previous studies have shown the intercomparison of dual Doppler wind fields from two orthogonal flight legs and a ground-based two-radar network [Jorgensen et al., 1983, Hildebrand and Mueller, 1985]. Several other studies have investigated both single and multi-

Doppler techniques for retrieving TC wind fields [Lee et al., 1994, Crum et al., 1998, Reasor et al., 2000, Lee et al., 1999, Jou et al., 2008, Bell et al., 2012], but the strengths and weaknesses of different techniques have not been compared and addressed fully. In this study, ground-based single Doppler and airborne dual Doppler observations simultaneously sampling Hurricane Matthew (2016) are analyzed to provide the first comprehensive comparison between ground-based single and airborne multi-Doppler wind retrieval techniques in a TC.

Line 27; I highly doubt that this is the first study to compare ground-based and airborne wind retrievals. The NOAA ground-based and airborne radars have been around for decades!

Thank you for the comment. We have added several references on these types of comparisons, and clarified that we mean specifically that the comparison between single-Doppler wind retrievals and airborne dual-Doppler has not been conducted. The comparison between ground-based dual Doppler and airborne wind retrievals was done in 1980s to 1990s, but the wind field retrieved by a single Doppler radar observation has not been compared with the airborne dual Doppler observations to our knowledge. Please refer to the previous response for the revised sentences and additional references.

Lines 56 – 57; sentence doesn't read right, "...only a small portion of TC"? Please rewrite.

Thank you for the comment. We have revised the sentence.

Ground-based dual-Doppler radar observations of TCs are usually limited to the observation of storms that happen to develop or move within the domain covered by the radars and extensive radar baselines [Jou et al., 1996].

General comment on the writing; at several places in the paper the word "the" needs to be inserted. Go through the paper again and look for these. Some examples:
Line 96, "...and P3 TDR..." needs a "the" before "P3"
Line 98, "...of KAMX radar..." needs a "the" before "KAMX"

Thank you for the comment. We have gone through the paper and added the missing word.

Line 122, Does this "mean" wind have the hurricane removed?

Thank you for the comment. The mean wind magnitude and direction were derived from the airborne dual Doppler analysis, following the procedure proposed by [Marks et al., 1992]. The storm-relative horizontal wind field $(V_r)$ in a cylindrical coordinate system centered on the storm can be decomposed into:

$$V_r(r, \theta, z) = \bar{V}_r(z) + V'(r, \theta, z) \tag{1}$$

where r is radius, $\theta$ is azimuth, z is height, $\bar{V}_r(z)$ is the horizontally averaged wind vector over the radius and azimuth, and $V'(r, \theta, z)$ is the deviation from $\bar{V}_r(z)$. $\bar{V}_r(z)$ can be expressed as:

$$\bar{V}_r(z) = \frac{1}{2\pi} \int_0^{2\pi} \int_0^{r_{max}} V_r(r, \theta, z) dr d\theta \tag{2}$$

If the horizontal wind field is from a circular symmetric vortex with no steering flow, $\bar{V}_r(z)$ would be zero. Nevertheless, if the vortex is embedded in the steering flow, the averaged horizontal wind field would equal the mean wind component. Thus, the local wind shear can be approximated by subtraction of the mean wind component at different altitudes. In our study, we calculated the mean wind component averaged from the vortex inner core area within the radius of 60 km.

The mean wind $(V_M)$ is the horizontal average of the environmental flow at each altitude following the procedure proposed by [Marks et al., 1992], which can be used to calculate the vertical wind shear.

Equation (4), The two angles in the second terms on the RHS of (4) should be THETAt, not THETA. This is probably just a typo.

Thank you for the comment. We have corrected the typoes.

$$Dcos\theta_d = Rcos\theta + R_Tcos\theta_T$$
$$Dsin\theta_d = Rsin\theta + R_Tsin\theta_T$$

(3)

Line 150, what is the lowest elevation angle used here and what error does this incur? The method must only work where cos(phi) $\sim$ 1, so lowest scan level only.

Thank you for the comment. Figure 2 shows the radar beam elevations of $0.5°$ and $0.9°$ versus the range. The distance between the KAMX radar and the TC center of the four radar volumes corresponding to the four flight passes are highlighted by the vertical lines. Our analysis focuses on the altitude of 4 km, and the four radar volumes at this altitude is constructed by the radar beam below $1°$ elevation angle. Therefore, the assumption of Vd $\sim$ Vd/cos($\phi$) is applicable.

Plugging Eq. 4 into Eq. 3 and approximating $\hat{V}_d/cos\phi$ with $V_d$ (only valid when the elevation angle is low):

Lines 177 – 179, this assumption is probably only valid above the boundary layer and below the outflow layer. Radial wind asymmetries can be substantial in the boundary layer. Some discussion of this is needed.

Thank you for the comment. We have added the conditions to the assumption.

This closure assumption may not be applicable within the boundary layer or outflow layer where the radial wind asymmetries can be substantial.

Page 8, this entire page could be significantly shortened because the "dynamic" centers are ultimately used, not the GBVTD centers. This can be summarized briefly.

Thank you for the comment. We would like to present the GVTD simplex algorithm which outperforms the GBVTD-simplex algorithm as a reference for future work. The spline fit of

[Figure]

Figure 2: The radar beam height of 0.5° and 0.9° elevation angles (black solid line). The distance between the KAMX radar and the TC center of the four radar volumes corresponding to the four flight passes are highlighted by the colored vertical lines. The dashed black line represents the altitude of 4 km.

the dynamic centers is sometimes quite variable, so we would want to provide the GVTD-simplex centers as another option to derive the TC centers with high temporal resolution in this manuscript.

Discussion regarding Figure 4 on page 9: I would say the results are mixed on the improvement of the "optimal solution" over the "original solution". For example, in one pass the green dashed line looks better than the blue dashed line, but in another pass, it looks worse and in the other passes the differences are negligible. Similar things for the orange and red lines.

Thank you for the comment. Since the four solutions are quite similar with only a few m/s apart, we have calculated the RMS difference results quantitatively in Table 3 to quantitatively demonstrate that the "optimal solution" outperforms other methods, and the RMS difference of the improved GVTD solution (red) is smaller than the original GVTD solution (orange). However, similar to the response to the major comment above, we have changed the tone in the manuscript to not overly state the results.

Also, on page 9: what are the heights of comparison between the TDR and the 88D? Since the storm core is on the far edge of the 88D coverage, the beam heights are probably fairly high, and the vertical velocities could be significant in this region.

Thank you for the comment. The altitude of the analyses are added in the manuscript.

The subsequent analyses use the observations at the altitude of 4 km, so the ground-based 0.5° radar elevation beam can detect the TC inner core.

What is the impact of significant vertical velocity in the hurricane core on the TDR and 88D comparisons, given that the 88D retrievals don't take this into account?

Eq. (2) in Lee et al. 1999 shows the vertical velocity and terminal velocity term (($w$-$v_t$)sin($\phi$)) in the construction of the Doppler velocity. They removed the contribution from $w$ and $v_t$, and divided cos($\phi$) to get Eq. (3). Therefore, the vertical velocity and terminal velocity contribution becomes ($w$-$v_t$)sin($\phi$)/cos($\phi$). sin($\phi$)/cos($\phi$) is approximate to 0 if we use the data mainly derived from the lowest scan ($\phi$=0.5, sin($\phi$)/cos($\phi = 8.72 \times 10^{-3}$, Fig. 2). Therefore, the vertical velocity and terminal velocity contribution can be neglected in this case. We have added the discussion to the manuscript:

Note that $\hat{V}_d$ neglects the contribution from the terminal velocity ($v_t$) and vertical velocity ($w$) (Eq. 2 in [Lee et al., 1999]). The contribution from $w$ and $v_t$ is small if the elevation angle of the radar beam is low (<1°).

Lines 265 – 268, These improvements are quite small, and I am wondering if they are statistically significant? I think the authors are overstating the impact of the improvements to the GVTD technique here and some rewording is needed.

Thank you for the comment. A nonparametric Wilcoxon signed-rank test has now been conducted to test the null hypothesis that two paired sets of the RMS differences derived from the original and improved GVTD algorithms are drawn from the same distribution. Results show that the RMS difference between the original and improved GVTD algorithm is statistically significant in both single and dual-Doppler analyses with 99% confidence.

A nonparametric Wilcoxon signed-rank test is conducted to test the null hypothesis that two paired sets of the RMS differences derived from the original and improved GVTD algorithms are drawn from the same distribution. The RMS difference between the original and improved GVTD algorithms is statistically significant with a $p$ value < 0.001 using both the projected dual-Doppler winds and single-Doppler velocities, indicating that we can reject the null hypothesis at the 1% significance level (99% confidence). The statistics suggest that the RMS differences distribution of wavenumber 0 tangential wind retrieved from the original GVTD algorithm are likely to be larger than those from the improved GVTD method.

Discussion around lines 295 – 296: these comparisons have significant differences between Ao and A1 coefficients in the eyewall region and it is not fair to say that they are "roughly consistent". Deviations of 2 m/s or less are a major error for Ao and A1 coefficients that have small values.

Thank you for the comment. We have revised the sentence.

The deviations of $A_0$, $A_1$ and $B_1$ coefficients between the two analyses within the eyewall

region (15 - 25 km) are less than 2 m s$^{-1}$.

Lines 303 – 308, please see major comment (1).

Thank you for the comment. Please refer to the response to major comment (1).

Table 3, The differences between the original and improved GVTD method are only 0.35 m/s and this difference is likely not statistically significant. See major comment (2).

Thank you for the comment. Please refer to the response to major comment (2) and comment "Lines 265 – 268", where we have changed the tone of the manuscript and conducted a statistical significane test.

Table 4, I don't understand the wavenumber magnitudes listed here. Why is the magnitude of WV0 so low? This should be azimuthal mean, correct? There is some confusion in the naming conventions listed in the table and the text that needs fixing.

Thank you for the comment. We have corrected the table caption to be more specific.

$V_dD/R_T$ harmonics coefficients amplitude (harmonics 0 to 3) retrieved from the single Doppler and dual Doppler analyses.

Figure 6, should label these figures with the corresponding physical wind components because it is hard to follow.

Thank you for the comment. The physical wind components are added.

(a) $A_0$ (to obtain $V_RC_0$ and $V_Mcos(\theta_T - \theta_M)$) (b) $A_1$ (to obtain $V_RC_0$) (c) $B_1$ (to obtain $V_TC_0$) (d) $A_2$ (to obtain $V_RC_0$ and $V_TS_1$) (e) $B_2$ (to obtain $V_TC_1$) (f) $A_3$ (to obtain $V_RC_0$ and $V_TS_2$) (g) $B_3$ (to obtain $V_TC_0$ and $V_TC_2$)

**References**

[Bell et al., 2012] Bell, M. M., Montgomery, M. T., and Lee, W.-C. (2012). An axisymmetric view of concentric eyewall evolution in Hurricane Rita (2005). *J. Atmos. Sci.*, 69(8):2414–2432.

[Crum et al., 1998] Crum, T. D., Saffle, R. E., and Wilson, J. W. (1998). An update on the nexrad program and future wsr-88d support to operations. *Wea. Forecasting*, 13(2):253 – 262.

[Hildebrand and Mueller, 1985] Hildebrand, P. H. and Mueller, C. K. (1985). Evaluation of meteorological airborne Doppler radar. Part I: Dual-Doppler analyses of air motions. *J. Atmos. Oceanic Technol.*, 2(3):362 – 380.

[Hildebrand et al., 1994] Hildebrand, P. H., Walther, C. A., Frush, C. L., Testud, J., and Baudin, F. (1994). The ELDORA/ASTRAIA airborne Doppler weather radar: goals, design, and first field tests. *Proceedings of the IEEE*, 82(12):1873–1890.

[Jorgensen et al., 1983] Jorgensen, D. P., Hildebrand, P. H., and Frush, C. L. (1983). Feasibility test of an airborne Pulse-Doppler meteorological radar. *J. Appl. Meteor. Climatol.*, 22(5):744 – 757.

[Jou et al., 1996] Jou, B. J.-D., Deng, S.-M., and Chang, B. (1996). Determination of typhoon center and radius of maximum wind by using Doppler radar. *Atmos. Sci*, 24:1–24.

[Jou et al., 2008] Jou, B. J.-D., Lee, W.-C., Liu, S.-P., and Kao, Y.-C. (2008). Generalized VTD retrieval of atmospheric vortex kinematic structure. Part I: Formulation and error analysis. *Mon. Wea. Rev.*, 136(3):995–1012.

[Koch et al., 1983] Koch, S. E., desJardins, M., and Kocin, P. J. (1983). An interactive Barnes objective map analysis scheme for use with satellite and conventional data. *J. Appl. Meteor. Climatol.*, 22(9):1487 – 1503.

[Lee et al., 1999] Lee, W.-C., Jou, B. J.-D., Chang, P.-L., and Deng, S.-M. (1999). Tropical cyclone kinematic structure retrieved from single-Doppler radar observations. Part I: Interpretation of Doppler velocity patterns and the GBVTD technique. *Mon. Wea. Rev.*, 127(10):2419–2439.

[Lee et al., 1994] Lee, W.-C., Marks, F. D., and Carbone, R. E. (1994). Velocity track display—A technique to extract real-time tropical cyclone circulations using a single airborne Doppler radar. *J. Atmos. Oceanic Technol.*, 11(2):337–356.

[Marks et al., 1992] Marks, F. D., Houze, R. A., and Gamache, J. F. (1992). Dual-aircraft investigation of the inner core of Hurricane Norbert. Part I: Kinematic structure. *J. Atmos. Sci.*, 49(11):919–942.

[Ray and Stephenson, 1990] Ray, P. S. and Stephenson, M. (1990). Assessment of the geometric and temporal errors associated with airborne Doppler radar measurements of a convective storm. *J. Atmos. Oceanic Technol.*, 7(2):206 – 217.

[Reasor et al., 2000] Reasor, P. D., Montgomery, M. T., Marks, F. D., and Gamache, J. F. (2000). Low-wavenumber structure and evolution of the hurricane inner core observed by airborne dual-Doppler radar. *Mon. Wea. Rev.*, 128(6):1653–1680.

---

## Referee Report (RR1)

The authors have done a good job on most of the revisions/responses and the additions are clear. Thank you for this work. However, I still have one major comment that should be addressed. I am recommending major revisions to address this comment.

I disagree about the choice of 1 km horizontal grid spacing used in the NOAA P3 retrievals. It doesn't make much sense to use grid spacing below the actual sampling of the radar because the energy will be significantly reduced here anyway. The authors can choose whatever grid spacing they want (even 50-meter spacing), but this doesn't mean anything because these scales are not sampled, and they will be severely damped in the retrievals. The best horizontal grid spacing one can get is limited by the beam sampling interval, which is ~ 1.4 km for the current data. NOAA HRD uses slightly larger grid spacing at 2 km because of this but could probably get away with ~1.4 km spacing. The Koch et al. paper is useful, but very old and it only addresses the influence of the weighting factor or influence radius in the data "resolution". New research has shown that the real "resolution" of wind retrieval methods is larger than that quoted by the authors (4dx) and has other contributions from the solution method (3DVAR), additional filtering sometimes used such as Laplacian filtering, post-processing and QC methods. In addition, the authors need to state in the paper the raw sampling of the P3 radials (~1.4 km) and the Gaussian filtering applied (4dx), which results in fully resolved fields at 5.6 km, which is really the best it can get. Currently, none of this is mentioned in the paper.

---

## Author Response (AR2)

amt-2020-240

Thank you for reviewing the manuscript and providing constructive comments. We have made edits to the manuscript incorporated with your suggestions. Reviewers' comments are shown in black, our response to each comment is shown in blue, and changes to the manuscript are shown in red.

Report #1:
Review of "Comparison of Single Doppler and Multiple Doppler Wind Retrievals in Hurricane Matthew (2016)"
General comments: I thank the authors for their thoughtful replies and careful revisions to their manuscript. The authors have addressed most of my comments adequately. I have comments on Eq. 20 that is newly derived in the revised manuscript.

Specific comments: L202: This is a comment. Eq. 20 describes that the GVTD technique cannot derive the axisymmetric radial wind at R=RT because of the singular point. This is one of the limitations in the GVTD technique, which is not described in Jou et al. (2008).

We have noted the limitations of deriving VRC0 when R = RT in the revised manuscript.

One caveat of the $V_R C_0$ updated form is that the axisymmetric radial wind cannot be derived when $R = R_T$ because of the singular point.

L356: When the single Doppler analysis is used, is your reply "the retrievals of A2, A3 are variable due to the propagation of wavenumber 2 winds" correct? As shown in Fig. 7b, the GVTD technique using the harmonic 2 and 3 components from the single Doppler analysis can retrieve wavenumber 2 tangential wind reasonably, which is consistent with linear vortex Rossby wave theory. The authors hypothesize in the next paragraph that the discrepancies of retrieved wavenumber 1 and 2 tangential winds are attributed to the dual Doppler wind synthesis. Thus, I think the GVTD technique can also retrieve the axisymmetric radial wind from the single Doppler analysis and it can be validated by using the wavenumber 0 radial wind retrieved by the dual Doppler analysis, which is reliable. However, I do not mean that the authors should do that in this study. I mean, the authors should describe that it is possible but that it is future work. As I wrote in the previous comment, information on the axisymmetric radial wind can be very useful for monitoring and predicting tropical cyclones.

We agree with the reviewer that the axisymmetric radial wind retrieval is valuable for monitoring and predicting tropical cyclones. The retrieval of axisymmetric radial wind could be reasonable when VRC2 and higher order terms are negligible (Eq. C1).

$$V_R C_0 = \frac{A_0 + A_1 + A_2 + A_3 + A_4}{(1 - \frac{R^2}{R_T{}^2})} - \frac{A_0 + A_2 + A_4}{(1 - \frac{R}{R_T})}$$
$$- V_R C_2 - \frac{\frac{R}{R_T}}{1 - \frac{R}{R_T}} V_R C_4 - \frac{1}{2}(\frac{1}{1 - \frac{R}{R_T}})(V_T S_5 - V_R C_5) \tag{C1}$$

However, our analysis shows that the propagation of wavenumber 2 tangential wind is aliased onto the steady wavenumber 1 component, resulting in a reduced amplitude and a phase shift in $A_2$ and $B_2$ in the dual Doppler analysis Moreover, [Lee et al., 2006] shows that the Lamb solution of VR2 has comparable magnitude as VT2 but with a phase shift, so the

wavenumber 0 radial wind retrieval is uncertain when VRWs are present.

Since the axisymmetric radial wind is influenced by the harmonics 2 and 3, and wavenumber 2 radial wind component (Eq. C1), we cannot fully validate the axisymmetric radial wind retrieval with the current dataset. [Lee et al., 2006] shows that the Lamb solution of the wavenumber 2 radial wind has comparable magnitude as the wavenumber 2 tangential wind but with a phase shift, so the wavenumber 0 radial wind retrieval is uncertain when VRWs are present. The evaluation for the accuracy of the axisymmetric radial wind retrieval is not included in this study.

Report #2:

The authors have done a good job on most of the revisions/responses and the additions are clear. Thank you for this work. However, I still have one major comment that should be addressed. I am recommending major revisions to address this comment. I disagree about the choice of 1 km horizontal grid spacing used in the NOAA P3 retrievals. It doesn't make much sense to use grid spacing below the actual sampling of the radar because the energy will be significantly reduced here anyway. The authors can choose whatever grid spacing they want (even 50-meter spacing), but this doesn't mean anything because these scales are not sampled, and they will be severely damped in the retrievals. The best horizontal grid spacing one can get is limited by the beam sampling interval, which is 1.4 km for the current data. NOAA HRD uses slightly larger grid spacing at 2 km because of this but could probably get away with 1.4 km spacing. The Koch et al. paper is useful, but very old and it only addresses the influence of the weighting factor or influence radius in the data "resolution". New research has shown that the real "resolution" of wind retrieval methods is larger than that quoted by the authors (4dx) and has other contributions from the solution method (3DVAR), additional filtering sometimes used such as Laplacian filtering, post-processing and QC methods. In addition, the authors need to state in the paper the raw sampling of the P3 radials ( 1.4 km) and the Gaussian filtering applied (4dx), which results in fully resolved fields at 5.6 km, which is really the best it can get. Currently, none of this is mentioned in the paper.

We have now added more clarification to the manuscript on our chosen analysis parameters based on the reviewers comment. One difference between SAMURAI and other analysis software is that 'grid-spacing' is not really an accurate term in this context since the software uses a finite element approach. SAMURAI employs cubic B-splines as a set of basis functions on the 'nodes' which can be used to represent any arbitrary function. The nodal spacing determines the minimum feature size resolved by the function, which is determined by sampling theory. We then apply the low-pass filter as part of the cost-function minimization, with the amount of filtering is specified by the user. In our study, the Gaussian recursive filter length was set to $4\Delta$ the nodal spacing in the horizontal and $2\Delta$ filter in the vertical. We set the nodal spacing ($\Delta$) to 1 km in the horizontal and 0.5 km in the vertical, which results in a 3D wind-field 'function' that can depict features with wavelengths larger than 4 km in the horizontal and 1 km in the vertical. We have chosen a 1-km nodal spacing with a $4\Delta$ filter, but could get a similar functional representation with a 0.5-km spacing and $8\Delta$ filter, or 2-km nodal spacing with a $2\Delta$ filter. As the reviewer correctly points out, the primary limitation for what features are actually resolved in the analysis depends on the data spacing. We have found through analytic testing that the features too close to the $2\Delta$ nodal scale may not be well-represented even with perfect data sampling. It is better to have a slightly finer nodal spacing with a little more filtering than nodal spacing that matches the data spacing exactly. With the along-track spacing $\sim$ 1.4 km, the best possible resolution of derived fields with any technique is 2.8 km, and is more accurately closer to 4 - 6 km depending on noise and details of the specific features and sampling. As such, we believe the minimum horizontal resolved spatial scale of our wind analysis at 4 km is sufficient, so that physical features larger than this scale can be well-represented by the spline function. We have added more of these details to the text to address the reviewer's concerns, as well as references to Ooyama (1987) and Ooyama (2002) which are relevant to the cubic B-spline representation of atmospheric structure.

The dual-Doppler analysis was synthesized with each of the P3 radial passes at 1-km horizontal spline nodal spacing and 0.5 km vertical nodal spacing using SAMURAI software

(Bell et al. 2012) in LROSE, with a 4$\Delta$x Gaussian filter in the horizontal and 2$\Delta$x filter in the vertical applied. SAMURAI is a three-dimensional variational data assimilation tool that uses a finite element basis to estimate the most likely state of the atmosphere given a set of observations. The nodal spacing of the finite elements should be smaller than the data spacing in order to accurately represent a spline function that can depict the spatial scales resolved by a given data sampling (e.g. Koch et al. 1983, Ooyama 1987, Ooyama 2002) . For the P-3 TDR in 2016, the data spacing is limited in the along-track direction to $\sim$ 1.4 km due to the rotation rate of the radar. With the chosen spline nodal spacing and Gaussian filter length the minimum resolved scale is $\sim$4 km in the horizontal, or approximately 2.85 times the along-track data spacing. Larger-scale features such as low azimuthal wavenumber structures are well-resolved by the analysis.

**References**

[Koch et al., 1983] Koch, S. E., desJardins, M., and Kocin, P. J. (1983). An interactive Barnes objective map analysis scheme for use with satellite and conventional data. *J. Appl. Meteor. Climatol.*, 22(9):1487 – 1503.

[Lee et al., 2006] Lee, W.-C., Harasti, P. R., Bell, M. M., Jou, B. J.-D., and Chang, M.-H. (2006). Doppler velocity signatures of idealized elliptical vortices. *TAO: Terrestrial, Atmospheric and Oceanic Sciences*, 17(2):429.

[Ooyama, 1987] Ooyama, K. V. (1987). Scale-controlled objective analysis. *Mon. Wea. Rev.*, 115(10):2479 – 2506.

[Ooyama, 2002] Ooyama, K. V. (2002). The cubic-spline transform method: Basic definitions and tests in a 1d single domain. *Mon. Wea. Rev.*, 130(10):2392 – 2415.